# A type VII-secreted lipase toxin with reverse domain arrangement

Stephen R. Garrett [1], Nicole Mietrach[1], Justin Deme [2], Alina Bitzer [3], Yaping Yang[1], Fatima R. Ulhuq [1], Dorothee Kretschmer [3], Simon Heilbronner[3,4], Terry K. Smith[5], Susan M. Lea [2] & Tracy Palmer [1] ✉

The type VII protein secretion system (T7SS) is found in many Gram-positive bacteria and in pathogenic mycobacteria. All T7SS substrate proteins described to date share a common helical domain architecture at the N-terminus that typically interacts with other helical partner proteins, forming a composite signal sequence for targeting to the T7SS. The C-terminal domains are functionally diverse and in Gram-positive bacteria such as *Staphylococcus aureus* often specify toxic anti-bacterial activity. Here we describe the first example of a class of T7 substrate, TslA, that has a reverse domain organisation. TslA is widely found across Bacillota including *Staphylococcus*, *Enterococcus* and *Listeria*. We show that the *S. aureus* TslA N-terminal domain is a phospholipase A with anti-staphylococcal activity that is neutralised by the immunity lipoprotein TilA. Two small helical partner proteins, TlaA1 and TlaA2 are essential for T7-dependent secretion of TslA and at least one of these interacts with the TslA C-terminal domain to form a helical stack. Cryo-EM analysis of purified TslA complexes indicate that they share structural similarity with canonical T7 substrates. Our findings suggest that the T7SS has the capacity to recognise a secretion signal present at either end of a substrate.

The ability to move proteins across biological membranes is a critical aspect of biology. The Sec and Tat pathways are found throughout prokaryotes and mediate transport across the cytoplasmic membrane. Targeting of globular proteins to either of these pathways is via a signal peptide that is located at the substrate N-terminus[1,2]. The type VII secretion system (T7SS), present in the diderm mycobacteria and in monoderm Gram-positives also exports proteins across the cytoplasmic membrane[3]. To date, all substrates of the T7SS share a common architecture; the N-terminal domain forms a helix-turn-helix domain (often containing a WXG or LXG motif), that is required for secretion and dimerisation, with a C-terminal functional domain of variable length[4].

The T7SS in mycobacteria and in Bacillota, such as *Staphylococcus aureus*, are only distantly related and have been designated T7SSa and

T7SSb, respectively[5]. The T7SSa, also termed ESX, can be found in up to five distinct copies in pathogenic mycobacteria. The ESX systems are heavily linked to virulence, with at least three of these being critical for host interaction and pathogenesis[e.g6–8]. By contrast, one of the primary roles of the T7SSb is interbacterial competition[9–11]. Regardless of biological function, all T7 systems depend on a membrane-bound FtsK/SpoIIIE family ATPase for activity. Termed EccC in the T7SSa, it forms a hexameric pore at the centre of a 2.3 MDa complex. Other subunits of the T7SSa are EccB, EccD, EccE and a periplasmically-located protease MycP[3,12]. The T7SSb ATPase, EssC, also forms hexamers, but interacts with a distinct set of partner proteins that differ in sequence and structure from those of the T7SSa[13–15].

The canonical T7SS substrates are proteins of the WXG100 family, comprising small helical hairpins that form homo- or heterodimers[16,17].

[1]Newcastle University Biosciences Institute, Newcastle University, Newcastle upon Tyne NE2 4HH, UK. [2]Center for Structural Biology, Center for Cancer Research, National Cancer Institute, NIH, Frederick, MD 21702, USA. [3]Interfaculty Institute of Microbiology and Infection Medicine, University of Tübingen, 72076 Tübingen, Germany. [4]German Center for Infection Research (DZIF), partner site Tübingen, Tübingen, Germany. [5]School of Biology, Biomedical Sciences Research Complex, University of St. Andrews, North Haugh, St. Andrews, United Kingdom. ✉e-mail: tracy.palmer@ncl.ac.uk

While these were originally identified as effector proteins[e.g18], it is becoming increasingly clear that at least some of them also serve as stabilising and/or targeting factors for larger T7SS substrates, with which they are co-secreted[10,19–21]. The LXG proteins are large antibacterial toxins secreted by the T7SSb, which form complexes with two or three WXG100-like partners. These small partners, which have been designated Lap (LXG-associated α-helical protein) interact with the N-terminal helical LXG domain and are predicted to form a helical stack[19–21]. Larger substrates of the T7SSa include the proline-glutamic acid (PE) and proline-proline-glutamic acid (PPE) proteins. These proteins heterodimerise through their N-terminal α-helical domains which also stack to form helical bundles[22]. Despite low sequence conservation between PE proteins and Laps there is striking structural similarity between PE25 and LapC1 suggesting commonalities in substrate organisation and targeting between the two types of T7SS[19].

In this study we describe a family of T7SS substrates that show the reverse structural organisation. We demonstrate that TslA is a T7SS-secreted interbacterial toxin from *S. aureus* which has a phospholipase domain at its N-terminus, and a helical C-terminal domain. We identify TlaA1 and TlaA2 as Lap-like proteins that interact with the TslA C-terminus to mediate its secretion. This defines a new class of T7SS effector where the targeting domain is found at the opposite terminus and indicates that the T7SS has the property of secreting proteins in both N-to-C and C-to-N directions.

## Results

### SAOUHSC_00406 (TslA) is a T7SS substrate encoded at a conserved genetic locus

Previous proteomic analysis of culture supernatants from a *S. aureus* wild type and isogenic *essC* mutant identified 17 proteins that showed greater than two-fold higher abundance in the wild type secretome, including the T7-secreted DNase toxin EsaD, its Lap-related partner EsxD and the LXG domain protein TspA[23]. Several other candidates were eliminated as T7 substrates, either because they were known cytoplasmic proteins, or shown to be secreted T7-independently. The singular exception was SAOUHSC_00406, which resulted in cell leakage when it was overexpressed with a Myc tag, preventing any conclusion about its usual subcellular location[23].

SAOUHSC_00406, which we renamed TslA for reasons outlined below, is encoded on the *S. aureus* vSaα island, alongside *set* exotoxin genes, two restriction endonucleases, and a variable cluster of DUF576-family tandem-like lipoproteins[24]. TslA is encoded at the 3′ end of the DUF576 gene cluster and precedes genes coding for two small helix-turn-helix proteins, SAOUHSC_00407 and SAOUHSC_00408 (Fig. 1a; subsequently renamed TlaA1 and TlaA2, respectively). Prior studies identified a promoter directly upstream of the tandem lipoprotein gene, *SAOUHSC_00405* (which we renamed *tilA*), and a transcriptional terminator directly after *tlaA2* indicating these genes are transcriptionally coupled[24,25]. This four gene cassette is also found in *Listeria* spp. and is embedded within the T7SS locus of some *Enterococcus faecalis* strains (Fig. 1b), consistent with a link to type VII secretion.

We noted that the N-terminal region of TslA was polymorphic across different *S. aureus* strains (Supplementary Fig. 1). Up to two further copies of this four gene locus can be found encoded at other locations on the *S. aureus* chromosome in a strain-dependent manner (Fig. 1c). In commonly studied strains such as USA300, NCTC8325 (the parental strain of RN6390) and COL, only one further TslA homologue is encoded, SAOUHSC_02786 (TslB), the gene for which carries a frame shift at codon 345 and it therefore does not align with the final approximately 100 amino acids of TslA (Supplementary Fig. 2a). However, in other strains such as ST398 the frameshift is absent, and the protein aligns with TslA and with a third homologue, CO08_0212 (TslC), encoded by strain CO-08, along its entire length (Supplementary Fig. 2b). Across *S. aureus* strains, TslA proteins share 75-94%

identity, with most of the sequence variability falling in the polymorphic N-terminal region. By contrast, in strain CO-08 which encodes full-length variants of all three Tsl1 proteins, TslA shares 47% identity with TslB and 42% identity with TslC, while TslB shares 47% identity with TslC. Analysis of the *S. aureus* genome sequences present in the RefSeq database indicated that 78.5% encode a full-length copy of TslA, 41.3% a full-length copy of TslB and only 13.3% a full-length copy of TslC (Supplementary Table 1).

To determine whether TslA is secreted by the T7SS, we used a sensitive assay based on Nanoluciferase binary technology (NanoBit[26,27]; Fig. 1d). This approach is more robust than western blotting which has been used previously to assess secretion because it is quantitative and avoids the requirement to concentrate supernatant proteins by precipitation[28]. Fusing pep86, the small fragment of nanoluciferase to EsxA, a T7SSb-secreted WXG100 protein that is a core component of the machinery, allows extracellular EsxA to be detected by supplementing the bacterial culture (Fig. 1e), or clarified supernatant (Supplementary Fig. 3a) with the large nanoluciferase fragment, 11 S. This reconstitutes enzyme activity and results in bioluminescence, the level of which is dependent on a functional T7SS[28]. We fused pep86 to the N-terminus of TslA and produced it in the *S. aureus* USA300 wild type strain and a Δ*essC* derivative. This resulted in only low levels of luminescence detected in the whole cell culture or supernatant of either strain (Fig. 1e, Supplementary Fig. 3b). However, when pep86-TslA was produced in tandem with TlaA1 and TlaA2 the extracellular luminescent signal was much higher from the wild type supernatant than the *essC* mutant. Control experiments indicated that the total cellular levels of pep86-tagged TslA was similar in the two strains (Supplementary Fig. 3c, Supplementary Fig. 4c), and that levels of a cytoplasmic protein, TrxA, tagged with pep86 were barely detected in the supernatant of either strain (Fig. 1e, Supplementary Fig. 3b, Supplementary Fig. 4b). When either *tlaA1* or *tlaA2* were absent from the construct, secretion of pep86-TslA was drastically reduced, indicating that both encoded proteins are required for the efficient secretion of TslA by the T7SS (Supplementary Fig. 3a,b, Supplementary Fig. 4a, b).

Previous studies have shown that some small Lap partner proteins are co-secreted with their cognate LXG toxin[e.g20] whereas others may be retained in the cytoplasm[19]. To determine whether TlaA1 or TlaA2 are also secreted by the T7SS, pep86 fusions were constructed to each and co-produced alongside the other small partner and TslA. The extracellular luminescent signal from either of these constructs was significantly greater in a strain where the T7SS was functional (Fig. 1e; supplementary Fig. 3b, Supplementary Fig. 4a, b) although it was higher for the TlaA2 construct than for TlaA1. Furthermore, when *tslA* was absent from these constructs, neither TlaA protein was secreted by the T7SS, confirming that TslA, TlaA and TlaA2 are all co-secreted (Supplementary Fig. 3a, b, Supplementary Fig. 4).

Taken together our results show that TslA is a T7SSb substrate and that TlaA1 and TlaA2 enhance TslA secretion. Based on these findings and other results presented below we have subsequently renamed SAOUHSC_00406 and its homologues as the Tsl1 family of lipases, and the small partner proteins linked to this family as Tla1 and Tla2, respectively. As TslA appears to be highly conserved at the vSaα island, we have named this TslA (Type VII secreted lipase A), with SAOUHSC_00407 as TlaA1 and SAOUHSC_00408 as TlaA2. The TslA homologue encoded at the LPLIII locus was renamed TslB and at the LPLI locus renamed TslC.

### TlaA1 and TlaA2 form a complex with the C-terminal 'LXG-like' domain of TslA

TslA is predicted to have two domains (Fig. 2a). Analysis using Phyre2[29] suggests that the N-terminus shares structural homology with a lipase from *Yarrowia lipolytica* (99.8% confidence modelled to PDB entry 3OOD[30] [https://www.rcsb.org/structure/3o0d]), while the C-terminal

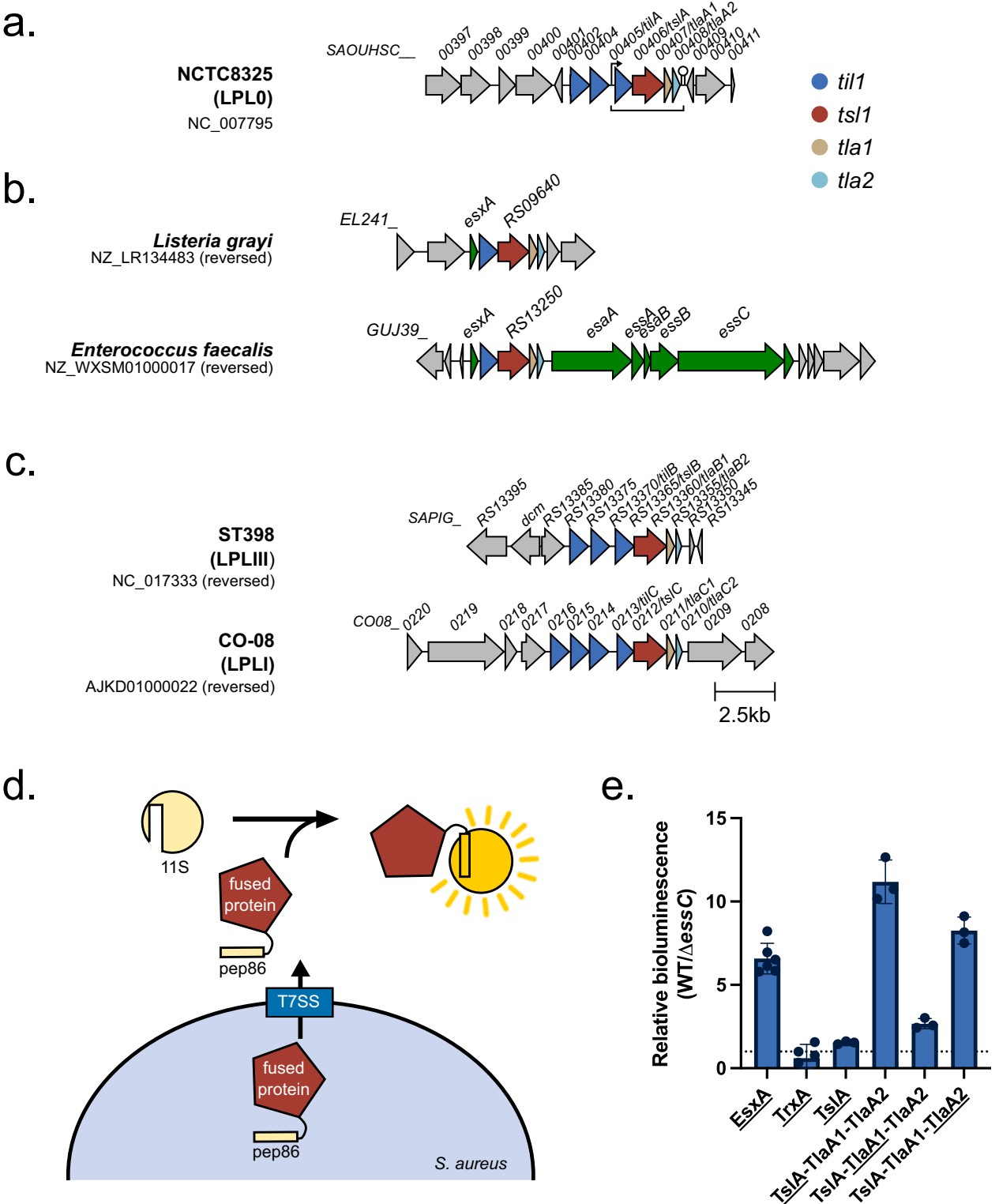

domain is predicted to be extensively α-helical and share structural homology with EsxA from *Geobacillus thermodenitrificans* (89.2% confidence modelled to PDB entry 3ZBH [https://www.rcsb.org/structure/3ZBH]). It therefore structurally resembles other T7SS substrates such as LXG proteins except that the domains show a reverse arrangement.

LXG domains are named after the conserved L-X-G motif found in the turn of the helix-turn-helix domain. Interestingly, the structural model of TslA predicts L312 and G310 to form a similar motif in the α-helical C-terminal LXG-like domain, however, this is formed by a G-X-L in the primary sequence (Fig. 2b). Alignment of the TslA model with the AlphaFold model of *Streptococcus intermedius* LXG protein TelC, shows close positioning of the L and G residues between the two proteins (Fig. 2b). To test whether this LXG-like motif is required for secretion of TslA, amino acid substitutions at positions G310 and L312 were introduced into pep86-tagged TslA, and the variant proteins produced alongside TlaA1 and TlaA2 in the split-nanoluciferase assay. From this we observed that although G310S or G310A had no detectable effect, the L312A substitution significantly impaired secretion of TslA (Supplemetary Fig. 3a, b, Supplementary Fig 4a, b).

**Fig. 1 | SAOUHSC_00406/TslA is encoded at a conserved gene cluster and secreted by the T7SS. a** SAOUHSC_00406/TslA is encoded on the vSaα island, at a locus also known as the LPL0 lipoprotein gene cluster. A variable number of SAOUHSC_00405/TilA homologues are encoded upstream of this cluster across different *S. aureus* strains, in RN6390 there are two (SAOUHSC_00402 and SAOUHSC_00404). **b** The four gene cassette is encoded in *Listeria grayi* and enterococcal genomes. **c** Homologues of *SAOUHSC_00406/tslA* can be found at a further two loci, LPLI and LPLIII, in a strain-dependent manner. Where strains do not encode a paralogue, the *tla1 (SAOUHSC_00407*-like) and *tla2 (SAOUHSC_00408*-like*)* genes are also absent, but variable numbers of *til1 (SAOUHSC_00405-like)* genes are present. To date no *SAOUHSC_00406* paralogue is found at the LPLII locus which always appears to encode a single, phylogenetically diverse Til1 protein[24] (see also Supplementary Fig. 7). **d** Schematic representation of the split nanoluciferase assay to detect T7SS-dependent secretion. **e** The pep86 fragment of nanoluciferase was fused to the indicated protein of interest (denoted with an underline) and expressed from plasmid pRAB11 in strain USA300 or an otherwise isogenic *essC* mutant as described in the methods. To 100 μl of whole cell samples, 11S fragment of nanoluciferase and furimazine were added, and luminescence readings taken over a 10 min time course. Peak readings were used to calculate relative luminescence of the wild type compared to the *essC* mutant strain as described in the methods. Data are presented as the mean ± SD (*n* = 3 biologically independent experiments). Readings for supernatant and cytoplasmic fractions from these samples are displayed in Supplementary Fig. 3a, b and the raw data from these experiments is shown in Supplementary Fig. 4. Source data are provided as a Source Data file.

Since LXG proteins interact with Lap partner proteins, we used AlphaFold to indicate whether TlaA1 and TlaA2 may directly interact with TslA. As shown in Fig. 2c, AlphaFold strongly predicted these two small proteins to bind to the helical LXG-like domain of TslA. To probe this further we first undertook bacterial two-hybrid analysis to examine interactions between these proteins (Supplementary Fig. 5) which confirmed a likely interaction between TlaA2 and the C-terminal domain of TslA. We next co-overproduced Strep-tagged TlaA1 and Myc-tagged TlaA2 alongside $TslA_{CT}$ which had been provided with a His tag. Following successive $Ni^{2+}$-affinity and Streptactin-affinity purification and size exclusion chromatography a complex of all three proteins could be isolated (Fig. 2d, e). We conclude that TlaA1 and TlaA2 form a tripartite complex with the C-terminal domain of TslA and that these three proteins are likely co-secreted as a complex by the T7SS.

## The N-terminal domain of TslA has phospholipase A1 and A2 activity

To determine if TslA has lipase activity, we overproduced and purified the full-length protein and the lipase-like domain in isolation (Supplementary Fig. 6a–d) and incubated the protein with the C12 fatty acid ester polyethylene glycol sorbitan monolaurate (Tween 20). Hydrolysis of Tween 20 by lipases releases the fatty acid, which will precipitate in the presence of $Ca^{2+}$, measured by an increase in $OD_{500}$[31]. Both constructs showed lipase activity in this assay (Supplementary Fig. 6e, f). Structural prediction of TslA indicates the presence of a predicted catalytic triad comprising Ser164, Asp224 and His251 (Fig. 2a, Supplementary Fig. 2b). Individual substitutions of each of these residues completely abolished lipase activity in the Tween 20 assay without affecting folding of the protein (Supplementary Fig. 6g, h).

Phospholipases are also secreted by other bacterial secretion systems[32–34]. Four different classes of type VI-secreted lipases, Tle1-4, have been characterised and all share the G-X-S-X-G motif that is also found at the active site of TslA. Tle1 and Tle2 are the best characterised of the four classes and have been shown to have phospholipase A1 ($PLA_1$) activity[34,35]. To determine whether TslA also has this activity we incubated the full-length protein or the catalytically inactive variants with the $PLA_1$ substrate, PED-A1. Figure 3a shows the release of fluorescence from this substrate catalysed by TslA that is lost upon inactivation of the active site. Activity was also observed against the $PLA_2$ substrate, PED6, suggesting that, similar to Tle1[35], TslA can cleave at either the *sn*-1 or *sn*-2 position (Fig. 3b). We conclude that TslA has phospholipase A activity.

## TslA interacts tightly with an inhibitory protein, SAOUHSC_00405/TilA

Bacterial secreted lipases often act as toxins targeting either eukaryotic hosts or the membranes of other bacteria. Antibacterial lipases are coproduced with immunity proteins that protect the producing bacteria from self-intoxication[34]. The first gene of the four gene *tslA*

cluster encodes a conserved DUF576-family lipoprotein that we have named TilA, for reasons outlined below. These proteins have been previously termed Csa (conserved staphylococcal antigen) or Lpl (Lipoprotein-like) and are encoded by multi-gene families found at four loci in *S. aureus* strains (Supplementary Fig. 7). They have been linked to the invasion of host cell keratinocytes and potent immune stimulation due to shedding from the *S. aureus* membrane[e.g.36–38]. However, at three of these four loci, TslA homologues can also be encoded, and we wondered whether their primary role was as TslA immunity proteins. This is also supported by the finding that DUF576 proteins are encoded in T7SS immunity islands[39] and they are found in highly variable copy number and sequence across *S. aureus* strains similar to the highly variable immunity repertoires for EsaD and TspA[40].

To probe this we first tested whether purified TilA lacking its signal sequence (Supplementary Fig. 8a) could interact with the lipase domain of TslA. Although only weak interaction could be detected by bacterial two hybrid analysis (Supplementary Fig. 5), isothermal titration calorimetry indicated that TilA binds to the TslA lipase domain with a 1:1 stoichiometry and a Kd of 14.2 nM (Fig. 3c). When TilA was titrated into the $PLA_1$ or $PLA_2$ assay, TslA activity was inhibited (Fig. 3d, Supplementary Fig. 8b). We therefore subsequently renamed SAOUHSC_00405 as TilA (Type VII Immunity against Lipase A).

We coproduced TilA without its signal sequence alongside TslA, TlaA1 and TlaA2 and could isolate a complex of all four proteins (Fig. 4a). Cryo-EM analysis of the purified complex yielded a 7.3 Å volume (Fig. 4b, c). Rigid body fitting of the TilA-TslA-TlaA1-TlaA2 AlphaFold model into the cryo-EM map shows gross conformational agreement between the map and model (Fig. 4b), though no density for TlaA1 was evident in the volume, indicating this subunit may have dissociated from the complex during vitrification. Only partial C-terminal density for TlaA2 was also observed. The three C-terminal TslA and two TlaA2 α-helices match the helical density of the cryo-EM volume and are apparent in 2D class averages (Fig. 4c) suggesting the positioning of the α-helices are consistent between the AlphaFold model and cryo-EM volume. In addition to the sample demonstrating preferred orientations in vitreous ice, it is probable the particles suffered from conformational heterogeneity which precluded higher resolution map reconstruction. Notably, the hinge region linking the globular N-terminal domain and helical C-terminal domain of TslA and interfacing residues are of lower confidence in the AlphaFold model which may indicate a degree of motion between both domains.

## TslA has antibacterial activity

The results described above point to a role for TslA as an antibacterial toxin. To confirm this, we tested whether TslA could self-intoxicate a *S. aureus* strain that lacks proteins of the TilA family. A USA300 mutant deleted for all 18 chromosomal copies of *tilA*-related genes has been constructed previously[41] (Supplementary Fig. 7b). This strain (herein named USA300 Δ*til1*) does not express *tslA* due to deletion of the start codon and additionally lacks the

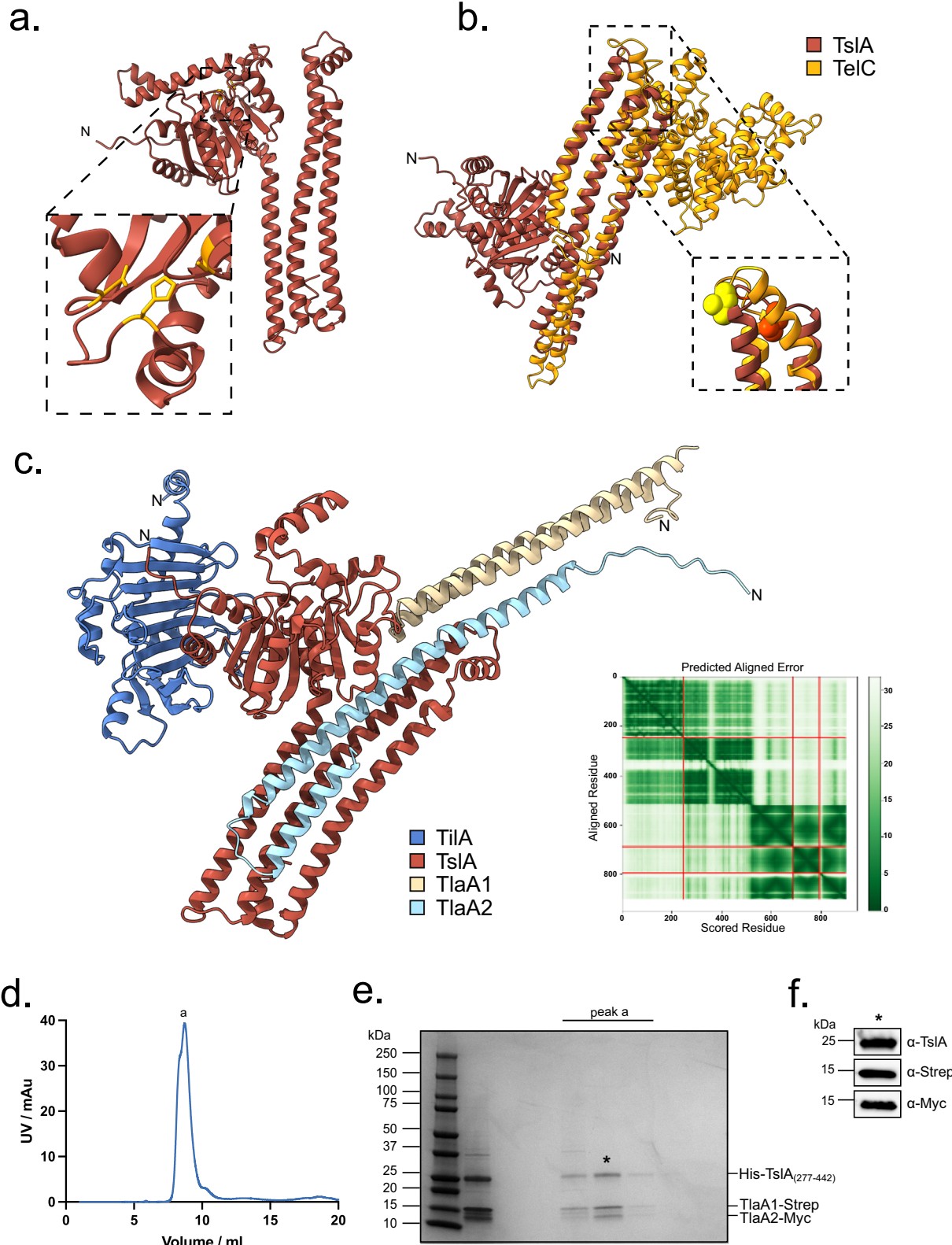

coding sequence for *SAOUHSC_02786/tslB*. The *til1* strain harbouring an empty plasmid vector (pRAB11) grows similarly to the wild type parent in TSB medium (Fig. 5a). However, although production of plasmid-encoded TslA alone did not affect either strain, co-production of TslA alongside secretion partners, TlaA1 and TlaA2, significantly slowed growth of the *til1* mutant (Fig. 5a). This growth inhibition was dependent on a functional T7SS because an otherwise

isogenic *til1* strain lacking the T7SS core component, EssC, but co-producing TslA-TlaA1-TlaA2 grew as well as the control strains (Fig. 5a). This confirms that TslA has extracellular toxicity and that it requires the T7SS for its secretion.

To confirm that toxicity is due to phospholipase activity, the active site substitutions S164A, D224A and H251A were individually introduced into TslA in the TslA-TlaA1-TlaA2 construct. None of these

**Fig. 2 | TslA has 'reverse' LXG architecture but can interact with Lap-like proteins. a** Structural model of TslA obtained from the AlphaFold Database[72]. The N-terminus is indicated. The inset shows the predicted active site. **b** The LXG-like C-terminus of TslA (maroon) aligned with the *S. intermedius* LXG protein, TelC (obtained from the AlphaFold Database and shown in gold). The inset depicts the L-X-G motif of TelC (yellow) and the G-X-L motif of TslA (red). **c** Model of the complex composed of TilA (blue), TslA (maroon), TlaA1 (beige), TlaA2 (pale blue) generated with AlphaFold Colab[75]. The predicted alignment error for the model is provided, with the sequence order being the same as the order listed above. **d** Size exclusion chromatogram of TslA$_{CT}$-TlaA1-TlaA2 containing fractions that had been previously co-purified by Ni-affinity chromatography followed by Streptactin affinity chromatography. This experiment has been performed three times, with similar results observed for each. AU−absorbance units. **e** SDS PAGE analysis of the indicated peak fractions from (**d**). **f** The protein fraction indicated with an asterisk in (**e**) was analysed by western blotting with anti-TslA, anti-Strep and anti-Myc antibodies, as indicated. This experiment has been performed twice, each with similar results. The uncropped blots from (**f**) can be found in Supplementary Fig. 10.

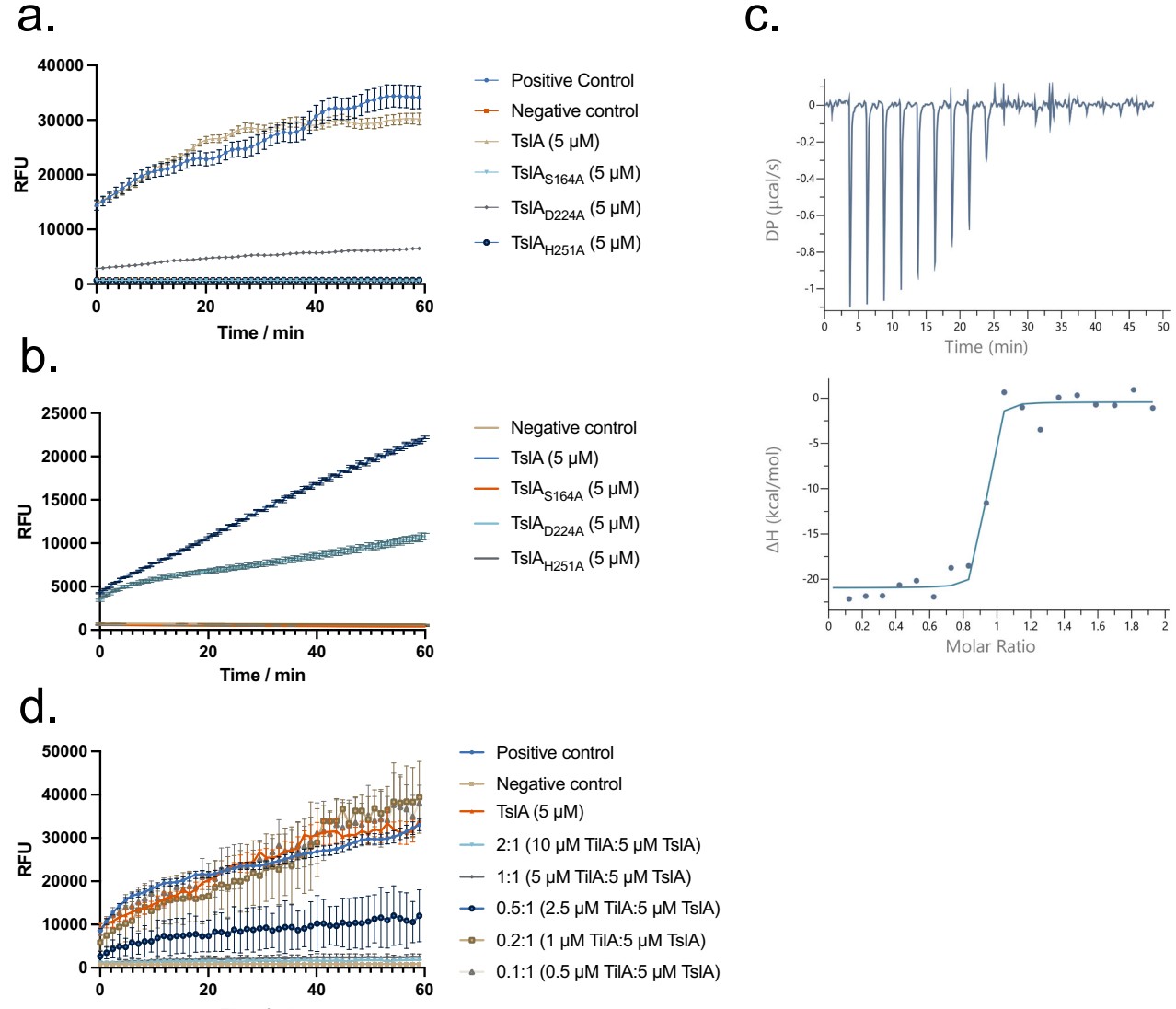

**Fig. 3 | TslA has phospholipase A$_1$ activity, which is inhibited by the immunity protein, TilA. a**, **b** Purified TslA and TslA with point mutants in the active site were incubated with (**a**) the PLA$_1$ substrate PED-A1 or (**b**) the PLA$_2$ substrate PED6. Fluorescence released upon substrate hydrolysis was measured at 515 nm over the course of 1 h. Data are presented as the mean ± SD. RFU−relative fluorescence units. **c** Calorimetric titration of TslA with TilA. (Upper) Raw data for the heat effect during titration. DP−differential power. (Lower) Binding isotherm. The best fit to the data gave $n = 0.88 \pm 0.01$ binding sites, ΔH = −20.5 ± 0.8 kcal mol$^{-1}$. **d** Hydrolysis of PED-A1 mediated by TslA alone, TilA alone or TslA and TilA at the indicated molar ratios. Negative control values were subtracted from each condition. The data presented in (**a**), (**b**) and (**d**) is $n = 3$ technical replicates. These experiments were repeated three times, with similar results observed each time. Error bars are ± SD. Source data are provided as a Source Data file.

variants were toxic (Fig. 5b) although they were produced and secreted to similar levels as wild-type TslA (Supplementary Fig. 3a, b, Supplementary Fig. 4, Supplementary Fig. 9).

Figure 5c demonstrates that the introduction of a copy of *tilA* onto the chromosome of the USA300 Δ*til1* strain was sufficient to fully protect the strain from TslA toxicity (Fig. 5c). Taken together the results in this section demonstrate that TslA has anti-staphylococcal activity which is rescued by the TilA immunity protein.

We next examined the membrane integrity of TslA-intoxicated strains using 3,3'-dipropylthiadicarbocyanine iodide (DiSC$_3$(5)) and

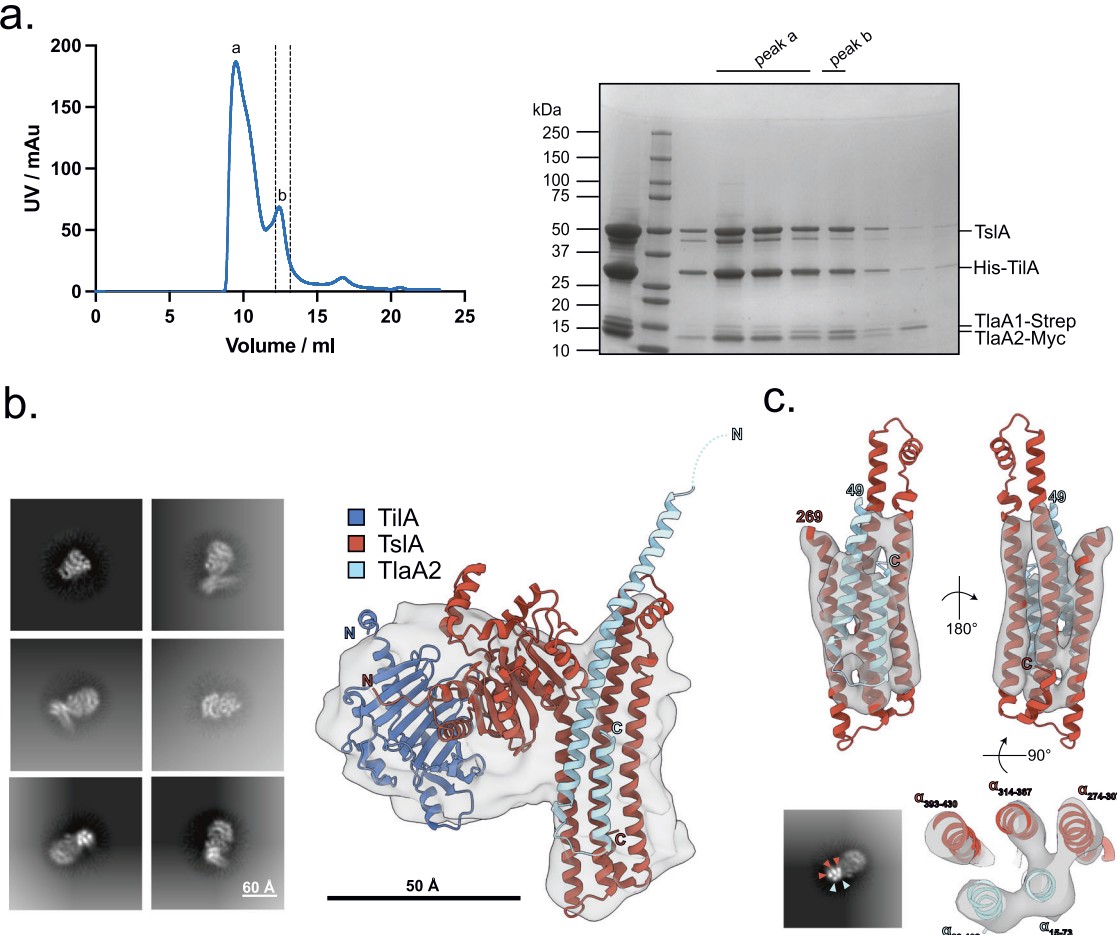

**Fig. 4 | TslA forms a complex with TilA, TlaA1 and TlaA2. a** Copurification of TilA, TslA, TlaA1 and TlaA2. (Left) SEC of the His-TilA-TslA-TlaA1-Strep-TlaA2-Myc complex following prior purification via HisTrap and StrepTrap affinity chromatography and (right) SDS PAGE analysis of peak fractions a and b from SEC. This experiment has been performed once. Peak fraction b was used for cryo-EM. AU—absorbance units. **b** 2D class averages (left) and 7.3 Å cryo-EM volume of TilA-TslA-TlaA2 complex (grey) with AlphaFold model docked into density at low contour level (right). **c** Side view of C-terminal TslA and TlaA2 α-helices docked into the same cryo-EM volume but at higher contour level (top). 2D class average demonstrating strong density for C-terminal TslA and TlaA2 α-helices coloured by subunit (bottom left) and slab view showing positioning of TslA and TlaA2 α-helices into cryo-EM density (bottom right). Residues corresponding to each α-helix are labelled.

Sytox green. DiSC$_3$(5) binds to hyperpolarised membranes and is integrated into the bilayer, staining only cells that have a membrane potential[23,42]. Sytox green binds to DNA, acting as a marker of significant membrane damage. Melittin, an amphipathic helical peptide from honeybee venom, was used as a positive control as it forms toroidal pores in membranes of bacteria including *S. aureus*[43,44]. Treatment of the wild-type strain with melittin induces depolarisation and DNA staining consistent with its action as a membrane-damaging antibiotic (Fig. 5d, e). However, production of TslA-TlaA1-TlaA2 in the same strain did not lead to any significant membrane damage or loss of membrane potential (Fig. 5d, e). In contrast, when TslA-TlaA1-TlaA2 were produced in USA300 Δ*til1*, a substantial number of cells became depolarised and those that did, were also stained by Sytox green (Fig. 5d, e; Supplementary Table 2), consistent with the action of TslA inducing membrane permeability.

To further characterise the membrane-damaging activity of TslA, total lipid extracts were prepared from USA300 and USA300 Δ*til1* producing TslA-TlaA1-TlaA2, at two and 6 h post-induction, and analysed by mass spectrometry lipidomics (Fig. 6). When compared to vector control, no degradation of lipids was observed in USA300 TslA-TlaA1-TlaA2 (Fig. 6a, b). However, in the lipid analysis of USA300 Δ*til1* producing TslA-TlaA1-TlaA2 increasing acyl cleavage from phosphatidylgycerol (PG), the major phospholipid in bacterial membranes, was

clearly evident, shown by the increasing amounts of both *lyso*-PG and free fatty acid at 2 and 6 h post-induction, respectively (Fig. 6c, d). Taken together, these findings are consistent with the biochemical activity of TslA described above, and we conclude that the cellular toxicity arises from membrane damage and detrimental effects caused by the detergent-like effects of copious amounts of either *lyso*-acyl phospholipids and/or the free fatty acid generated through the action of TslA.

## TslA is not required for virulence in a mouse abscess infection model

Previous studies have reported a role for the *S. aureus* T7SS in murine abscess models of infection[45–48]. A strain lacking the *til1* cluster along with the *tslA* cassette encoded on the vSaα island of USA300 also led to a significant reduction in bacterial burden in a mouse skin-infection model[37]. The Til1 lipoproteins are known to be shed from the surface of wild-type USA300 where they promote immune stimulation, which presumably accounts for the reduced virulence seen when the encoding genes are deleted[36,37]. We therefore investigated whether TslA contributed to virulence of *S. aureus* in the skin abscess model. As shown in Fig. 7, inactivation of the T7SS by deletion of *essC* resulted in a strong reduction in bacterial burden compared with the wild-type USA300 strain. A strain lacking all *til1* genes was also significantly less

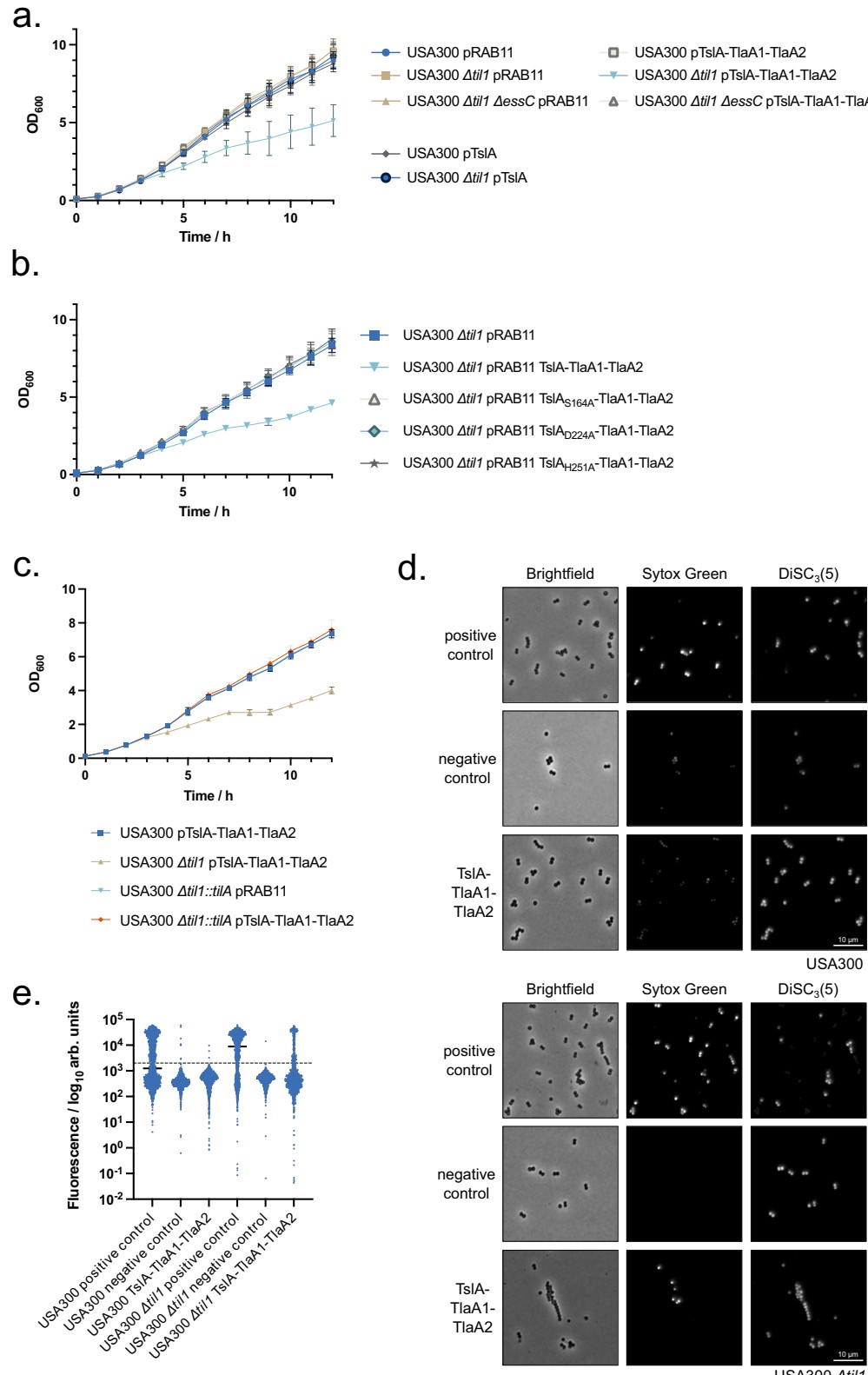

virulent than the wild type, in agreement with prior observations[37]. However, no statistically significant difference in bacterial burden was observed between USA300 and USA300 Δ*tslA*, although there was some trend towards decreased virulence of the *tslA* mutant in this infection model. (Fig. 7). It is possible that the virulence defect observed with the *essC* mutant is a cumulative effect resulting from the lack of secretion of TslA alongside other T7SS toxins.

## Discussion

In this work, we have identified TslA as a T7SS substrate protein. Previously, all characterised substrates of the T7SSa and T7SSb have helical targeting domains at their N-termini, often with C-terminal globular domains involved in nutrient uptake or having hydrolytic activity[9,10,48]. We show here that TslA instead has a globular N-terminal domain with phospholipase A activity, preceding the helical targeting domain at the

**Fig. 5 | TslA causes membrane damage to *S. aureus* cells in the absence of Til1 immunity proteins, in a T7SS-dependent manner. a–c** the indicated strain and plasmid combinations were cultured for 2 h, after which 500 ng ml$^{-1}$ ATc was added to induce plasmid-encoded gene expression. $OD_{600}$ readings, in triplicate, were taken manually every hour. The experiment was repeated three times and each point is an average of 3 biological and 3 technical replicates. Data are presented as mean values ± SD and $n = 3$ independent biological experiments. **d.** USA300 and USA300 Δ*til1* harbouring pRAB11 or pRAB11 encoding TslA-TlaA1-TlaA2 were cultured with ATc for 1 h 50 min, after which an aliquot of USA300/pRAB11 and USA300 Δ*til1*/pRAB11 were treated with 10 µM melittin for 5 min. Subsequently, all samples were stained with 200 nM Sytox green and 2 µM $DiSC_3(5)$, spotted onto an agarose pad and imaged by fluorescence microscopy. Representative fields of view, avoiding large clusters of cells are displayed. This experiment was repeated three times, each with similar results. Source data are provided on Figshare (https://doi.org/10.6084/m9.figshare.24648513). **e.** Fluorescence intensity of Sytox green for individual cells from each group plotted on a $log_{10}$ axis. At least 100 cells were analysed for each condition, pooled across three biological replicates. The exact number of cells analysed is provided in the Source Data file. Note that due to signal saturation in some fields, quantitative analysis could not be performed. Cells are therefore placed into high and low groups based on negative control values, depicted by the dashed line. The percentage of cells in each group with membrane damage is given in Supplementary Table 2.

C-terminus. TslA lacks any of the motifs, for example, WXG, LXG, PE or PPE that are associated with other T7SS substrates and would not be identified with algorithms that detect these. We anticipate that there are other reverse substrates yet to be uncovered, but that finding them may require structure-based searches, for example, to identify proteins with similar patterns of globular and helical domains.

The LXG toxins of the T7SSb interact with two or three small helical partner proteins, termed Laps, to form pre-secretion complexes and at least one of those Lap partners carries a C-terminal F/YxxxD/E motif required for LXG protein secretion[19]. A similar C-terminal motif is required for substrate secretion by the T7SSa and interacts with the EccC ATPase component[49–51]. Similarly, TslA requires two small helical partners, TlaA1 and TlaA2 for its secretion, which bind to the helical TslA C-terminus. However, there is no clear F/YxxxD/E or D/ExxxF/Y motif at either terminus of TlaA1 or TlaA2 and further work would be needed to dissect the features of these proteins required for targeting to the T7SSb.

Our results indicate that the primary function of TslA is as an antibacterial toxin. Surprisingly, however, despite being a phospholipase A, TslA is only toxic when it is exported outside the cell. This phenomenon has also been observed with the T6SS lipase substrate, Tle1, from *E. coli*, and with the T7SS lipid II phosphatase substrate, TelC[10,35]. TelC is dependent on $Ca^{2+}$ ions for activity, as are many lipases including most previously characterised staphylococcal enzymes[10,52]. Calcium is abundant in the cell wall of staphylococci but is buffered in the cytoplasm at low micromolar levels[53,54] and we speculate that the concentration is too low to support intracellular TslA activity. In support of an extracellular activity for TslA, *S. aureus* co-produces an extracellular immunity lipoprotein, TilA for self-protection, which binds to TslA with a low nanomolar Kd.

The organisation of gene clusters encoding LXG toxins often share a similar arrangement, with the Lap-encoding genes at the 5′ end of the cluster and the immunity gene at the 3′ end, directly adjacent to the toxin domain-encoding region[55,56]. It is striking that the entire *tslA* locus shows a reverse arrangement, with *lap*-like genes *tlaA1* and *tlaA2* at the 3′ end and the immunity gene at the 5′ end of the cluster but retaining the positioning immediately adjacent to the toxin domain. Another shared feature between TslA and the other *S. aureus* T7SS toxins is that the toxin domain is polymorphic across different *S. aureus* strains[9,23]. This may explain why multiple, non-identical copies of *tilA*-related genes are encoded at the *tslA* locus, to provide protection from TslA sequence variants found in other strains. Moreover, *S. aureus* strains can encode up to two further paralogues of the toxin, TslB and TslC, suggesting that this phospholipase is a key component of the *S. aureus* T7SS toxin armoury. It was noted that the *til1* immunity gene families are found in highly variable copy number, undergoing rapid copy number diversification during passage in vitro and in vivo[41]. They also undergo sequence shuffling by recombination across a highly conserved stretch of approximately 130 nucleotides in the central regions of the genes[24]. Similar features are also seen with immunity protein genes for the EsaD toxin in *S. aureus* and the TelC toxin in *Streptococcus intermedius*[40,51]. Til1 copies are encoded in the immunity gene islands of human commensal staphylococci such as *Staphylococcus warneri*[39] suggesting that TslA-mediated interbacterial competition is a likely feature of human skin and mucosal surfaces.

In conclusion, we have described an example of a T7SS substrate with a reverse domain arrangement, highlighting the unusual property of a prokaryotic cytoplasmic membrane transport system with the ability to recognise targeting domains at either end of a protein. In future it will be interesting to identify further examples that share this domain organisation and to determine how such substrates are recognised by the T7SS machinery.

## Methods
This research complies with all relevant ethical regulations. The murine study protocol was approved by "Regierungspräsidium Tübingen" of the University of Tübingen under the application number IMIT01/20G.

### Bacterial strains and growth conditions
*S. aureus* strains used in this study are listed in Supplementary Table 3. *S. aureus* was grown at 37°C, unless stated otherwise, on tryptic soy agar (TSA) or in tryptic soy broth (TSB) with vigorous agitation. Where required, media was supplemented with 10 µg ml$^{-1}$ chloramphenicol (Cml) for plasmid maintenance. Anhydrotetracycline (ATc) was used as counterselection for allelic exchange with pIMAY and pTH100 derivatives (100 ng ml$^{-1}$) and to induce expression from pRAB11 (500 ng ml$^{-1}$).

*E. coli* strains used in this study are listed in Supplementary Table 3. *E. coli* was grown at 37°C, unless otherwise stated, on lysogeny broth agar, or in lysogeny broth (LB) with vigorous agitation. Where required, media was supplemented with ampicillin (Amp, 100 µg ml$^{-1}$), kanamycin (Kan, 50 µg ml$^{-1}$) or Cml (25 µg ml$^{-1}$) for plasmid maintenance.

### Strain and plasmid construction
Plasmids were constructed by restriction digest, Gibson assembly or by direct synthesis from GenScript. Mutations in plasmids were introduced through site-directed mutagenesis, by PCR amplification and subsequent ligation by T4 ligase and polynucleotide kinase. Plasmids used in this study are described in Supplementary Table 4, with the primers used to construct them listed in Supplementary Data 1.

Isogenic mutants were constructed by allelic exchange using pIMAY, pIMAY-Z or pTH100, as described in publications[57–59]. For gene deletions, the upstream and downstream regions, including the first codon and final six codons for the gene to be deleted, were amplified from RN6390 genomic DNA. Cloning steps were carried out in *E. coli*, and following verification by DNA sequencing, were introduced into *S. aureus* strains by electroporation. Chromosomal deletion mutants were verified by amplification of the genomic region from isolated genomic DNA (GeneElute Bacterial Genomic DNA Kit, Sigma Aldrich) and by whole genome sequencing (MicrobesNG/Plasmidsaurus). To introduce a copy of *tilA* onto the chromosome, plasmid pTH100 was modified to replace *gfp* with *tilA*, which was confirmed by sequencing of the modified region. Chromosomal integration was performed as described[59].

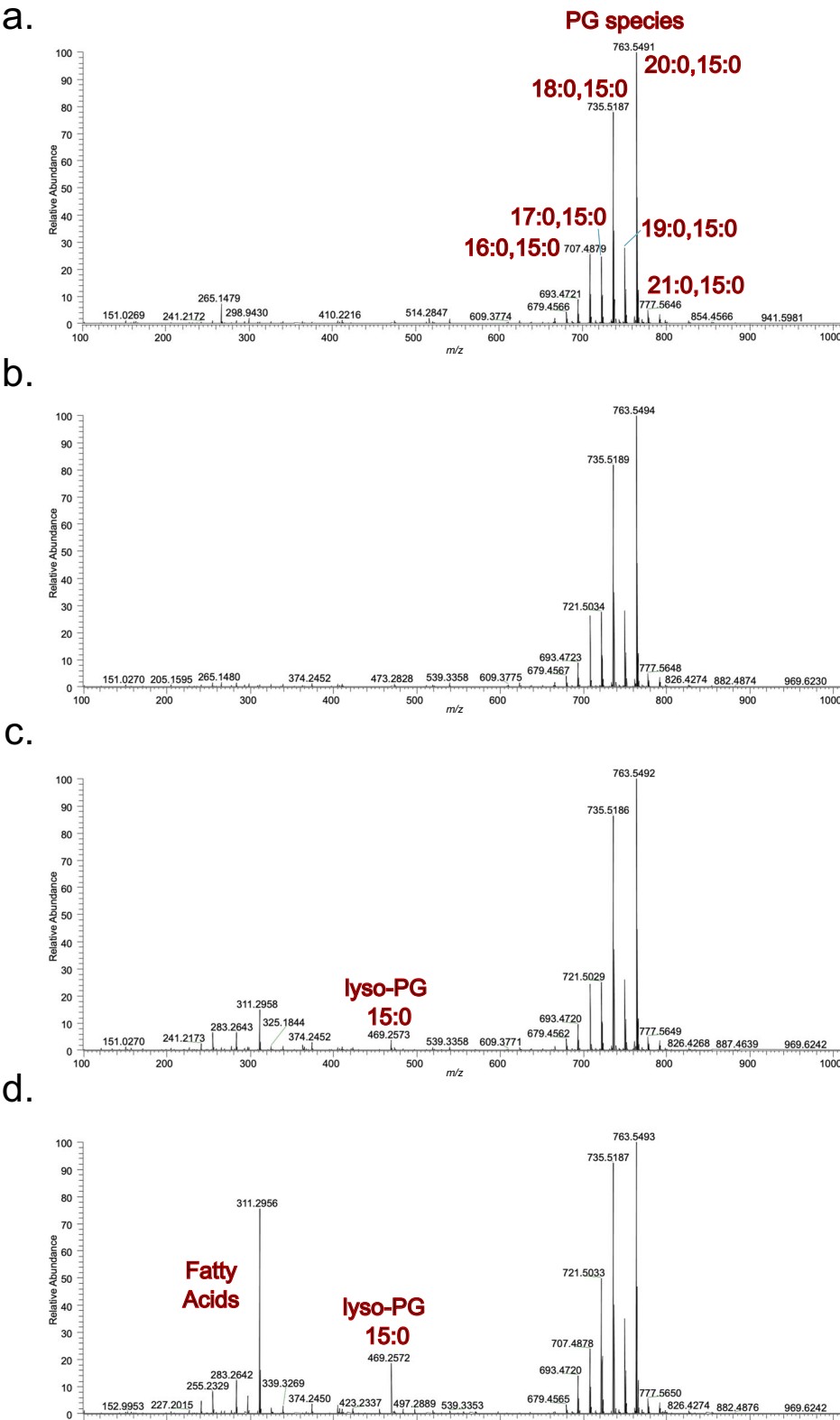

**Fig. 6 | Analysis of *S. aureus* membrane lipids following intoxication by TslA.** USA300 and USA300 Δ*til1* harbouring pRAB11 encoding TslA-TlaA1-TlaA2, were cultured for 2 h after which 500 ng ml⁻¹ ATc was added to induce plasmid-encoded gene expression. Samples were subsequently withdrawn at 2 and 6 h post-induction and membranes prepared as described in methods. Mass spectrometric analysis, in negative ion mode (100-1000 m/z) was carried out on membranes for **a.** USA300 pRAB11 and **b.** USA300 TslA-TlaA1-TlaA2 after 2 h post-induction, **c.** USA300 Δ*til1* TslA-TlaA1-TlaA2 after 2 h post-induction and **d.** USA300 Δ*til1* TslA-TlaA1-TlaA2 after 6 h post induction.

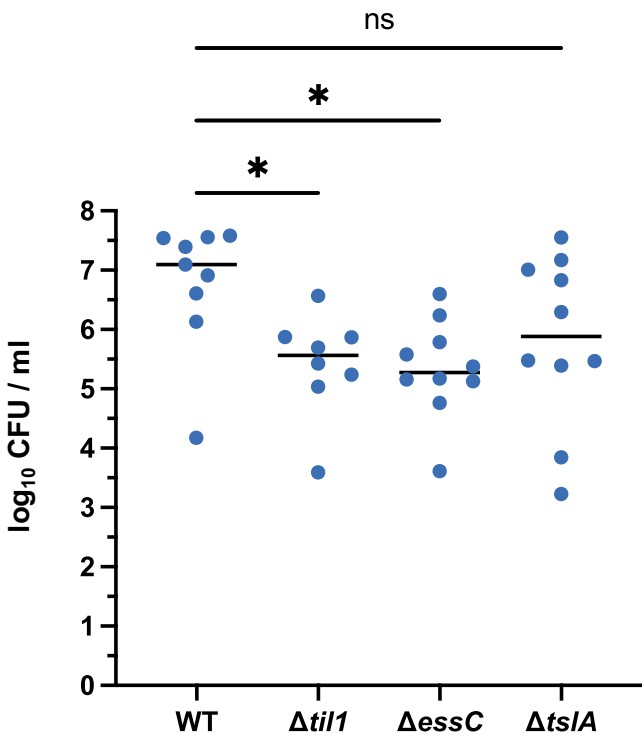

**Fig. 7 | TslA does not play a significant role in virulence in a murine skin abscess model.** Skin abscesses were induced by mixing $5 \times 10^4$ cells of each indicated strain with an equal volume of cytodex beads and inoculating into the flanks of mice. Abscesses were excised after 48 h, and c.f.u were enumerated. One-way ANOVA, assuming Gaussian distribution and equal SD was used to determine statistical significance. Multiple comparison of each strain against the wild type was used to demonstrate deviation from the control (n.s. $p > 0.05$; *$p < 0.05$). The $p$-values are as follows for wild type v.: $til1$ = 0.0392; $essC$ = 0.0197; $tslA$ = 0.1622. $n$ = 10 abscesses per condition. CFU – colony forming units. Source data are provided as a Source Data file.

## Split-nanoluciferase secretion assay

The split-nanoluciferase secretion assay was performed essentially as described previously[28]. Briefly constructs encoding the pep86 small nanoluciferase fragment fused to proteins of interest were assembled in pRAB11 (Supplementary Table 4) and introduced into *S. aureus*. Strains were grown overnight and subcultured in fresh TSB containing Cml to a starting $OD_{600}$ = 0.1. Cells were grown at 37 °C with shaking for 2 h before induction of pep86 fusion production by the addition of 500 ng ml$^{-1}$ ATc. Cells were grown for a further 2 h to an $OD_{600}$ = ~2. Triplicate 100 µl samples were withdrawn and taken as the whole cell culture. To obtain the cytoplasmic fraction, the equivalent of 1 ml of the sample of $OD_{600}$ = 1 was withdrawn, in triplicate, for each sample and pelleted by centrifugation. The top 100 µl of supernatant following pelleting was withdrawn and kept as the supernatant fraction. The cell pellet was suspended in 1 ml TBS supplemented with 2 mg ml$^{-1}$ lysostaphin (ambi) and incubated at 37 °C for 45 min. The cell samples were then boiled for 10 min to fully lyse cells. Samples were cooled to room temperature and serially diluted 1 in 2, in TBS, to 2$^{-3}$. 100 µl of each sample, in triplicate, were aliquoted into a Greiner CELLSTAR® white 96-well plate. To this, 5 µM 11S and 2 µl furimazine solution (Promega Cat. # N1610) were added and the luminescence read at 1 min intervals for 10 min using the FLUOstar Omega using a gain value of 3000. Data were analysed to find the time point which gave the highest luminescence value for the wild-type strain and this was divided by the luminescence value for the Δ*essC* strain at the same time point, for each condition, to give the relative luminescence[21]. The raw data used to calculate these relative values can be found in Supplementary Fig. S4. A Two-way ANOVA was performed, using multiple

comparison between groups to determine statistical significance (n.s. $p > 0.05$; *$p < 0.05$; ***$p < 0.001$; ****$p < 0.0001$).

## Bacterial two-hybrid analysis

Plasmids encoding T18 and T25 fusions were co-transformed into *E. coli* BTH101[60] and plated on MacConkey agar supplemented with 1% maltose and the required antibiotics and cultured at 30 °C for 40 h. Two colonies were picked in duplicate and separately cultured in LB containing Cml Amp overnight at 37 °C. Subsequently, 5 µl of these overnight cultures were spotted on MacConkey agar supplemented with 1% maltose and the required antibiotics and incubated for 40 h at 30 °C, after which they were imaged.

## Bacterial growth curves to assess TslA toxicity

*S. aureus* strains harbouring plasmids of interest were grown overnight and subcultured into 35 ml fresh TSB containing Cml 10 µg ml$^{-1}$ and supplemented with 5 mM CaCl$_2$, to an $OD_{600}$ of 0.1. Samples were taken at 0, 1 h and 2 h timepoints and $OD_{600}$ determined. After 2 h, all cultures were supplemented with 500 ng ml$^{-1}$ ATc. Subsequent $OD_{600}$ readings were taken every hour for a further 10 h. Each $OD_{600}$ measurement was taken in triplicate for each sample. Three biological replicates were performed for each experiment.

## Microscopy analysis

*S. aureus* strains USA300 and USA300 Δ*til1* harbouring pRAB11 or pRAB11_TslA-TlaA1-TlaA2 were grown overnight and subcultured to $OD_{600}$ 0.1 in fresh TSB containing Cml. Cultures were grown for 2 h before the addition of 500 ng ml$^{-1}$ ATc and were then incubated for a further 1 h 50 min at 37 °C with shaking. Cultures were diluted 1/3 in TSB and vortexed for 5 s. For strains harbouring the pRAB11 empty vector, two 200 µl aliquots were transferred into a 1.5 ml Eppendorf, while a single 200 µl aliquot was taken for the same strains expressing TslA-TlaA1-TlaA2. Melittin was added to a final concentration of 10 µM to one aliquot of USA300 and USA300 Δ*til1* pRAB11 cells (a positive control for cell lysis) and all aliquots were incubated at 37 °C for a further 5 min. To all samples, 2 µM DiSC$_3$(5) and 200 nM Sytox green were added and further incubated for 5 min. Following incubation, 1.5 µl of each cell suspension was spotted on a Teflon-coated multi-spot microscope slides (ThermoFisher) covered in a thin 1% agarose pad, dried and coverslip applied. Microscopy was performed using the Nikon Plan Apo ×100/1.40 NA Oil Ph3 objective, and a photometrics prime BSI sCMOS camera. The CoolLed pE-4000 LED light source was used, with either a Chroma 49002 and 460 nm LED for Sytox green or a Semrock Cy5-4040C with 635 nm LED for DiSC$_3$(5). Images were taken for five fields of view for each sample, for each of three biological replicates.

Images were analysed in a semi-automated way using Fiji[61]. All macros used are available at: https://github.com/NCL-ImageAnalysis/General_Fiji_Macros. Images were cropped to ensure even illumination across the field of view. The background fluorescence was determined by measuring the mean fluorescence intensity of a number of empty regions within each field of view and then subtracted from all pixels. Cells were identified using thresholding based upon phase contrast using ImageJ default thresholding. Selection was limited to particles of size between 0.423 µm$^2$ and 2.11 µm$^2$. Where two cells or a short chain were identified, these were manually separated by a 0.13 µm line. Large clusters of cells were excluded from the analysis, as were images with poor focus. Between 2-5 images for each condition were used from each of the 3 repeats. The fluorescence intensity of regions of interested was collected and collated for each condition. Quantitative analysis could not be performed for Sytox green staining as intensity was set using negative controls, resulting in some cells exceeding the maximum intensity. Cells were split into either positive or negative groups based on fluorescence intensity, with the fluorescence threshold based on the negative control. (e.g. greater than 2000 au for Sytox green was defined as positive and less than 2000 au as negative). The total in each group

was enumerated and the percentage of cells that were permeabilised based on Sytox green staining were calculated.

## Protein purification

For all protein overexpression work, overnight cultures were used to inoculate LB medium at a 1/100 dilution. Cultures were incubated at 37 °C with shaking (200 rpm) until $OD_{600} \sim 0.5$, after which they were supplemented with 1 mM isopropyl β-d-1-thiogalactopyranoside (or 0.2% $L$-arabinose for expression of 11S). Cells were cultured for a further 2–4 h depending on the expression construct (see Supplementary Table 5) after which they were harvested by centrifugation at $4000\,g$ washed with 1 X PBS and resuspended in the appropriate Buffer A (see Supplementary Table 5) supplemented with a cOmplete™ protease inhibitor cocktail tablet. Cells were lysed by sonication, debris were pelleted by centrifugation at 50,000 $g$ 4 °C for 30 min and the clarified supernatant was loaded onto the appropriate pre-equilibrated affinity chromatography column (Supplementary Table 5). After sample application, the column was washed with Buffer A until the $A_{280nm}$ reached baseline, after which the protein was eluted (see Supplementary Table 5 for elution buffers). The peak fraction was collected and concentrated using a 10 kDa spin concentrator. The sample was further purified by size exclusion chromatography (SEC) using Buffer B (see Supplementary Table 5 for column and buffer details). The peak fractions following SEC were analysed via SDS-Page and concentrated for further use. For purification of the TilA-TslA-TlaA1-TlaA2 complex this involved a further step – here the cell lysate was first applied onto 1 ml HisTrap column, with the peak fraction immediately applied to a StrepTrap column before the eluted fractions were further purified by SEC. Protein mass spectrometry was carried out as a service by the Metabolomics and Proteomics lab at the University of York.

The N-terminal His(6)tag was removed from purified TilA before the protein was used for further analysis. To achieve this, the peak fractions following elution from the HisTrap column were collected, pooled and TEV protease was applied to the sample in a 1:10 (w/w) ratio of protease:protein. TEV cleavage was performed overnight in a dialysis bag, while dialysing the sample in the appropriate Buffer A (Supplementary Table 5) at 4°C. The sample was separated from the TEV protease, which contains a His(6)tag by using a HisTrap column. The flow through, containing cleaved TilA, was collected and further purified by SEC.

## Protein electrophoresis and Western blotting

Protein samples were prepared for SDS PAGE by diluting 1:1 with 2 X Laemmli buffer and boiling for 10 min. Samples were microcentrifuged at full speed for 2 min before loading on a 4-20% SDS PAGE gel. Electrophoresis was carried out at 100 V for 10 min followed by 200 V for 40 min. For visualisation of total protein, gels were stained with Coomassie instant blue. For immunoblotting, proteins were transferred onto a nitrocellulose membrane using a Trans-Blot (BioRad), with Whatman paper soaked in Transfer buffer, composed of 25 mM Tris, 192 mM glycine pH 8.3, 10% Methanol. Polyclonal antibodies against TslA were raised in rabbits by Davids Biotechnologie, using purified TslA-His (Supplementary Fig. 6b). Following verification that the antibody recognised purified TslA, it was subsequently used at 1/1000 dilution, alongside an HRP-linked goat anti-rabbit secondary antibody (BioRad, catalogue number 1721019). Monoclonal antibodies against the Myc (Abcam, catalogue number ab23) and Strep (Qiagen, catalogue number 34850) tags were used at 1/5000 dilution, alongside an HRP-linked goat anti-mouse secondary antibody (BioRad, catalogue number 1721011). All uncropped Western blots are included in Supplementary Fig. 10.

## Circular dichroism

Proteins of interest were diluted to 0.5 mg ml$^{-1}$ in 20 mM NaPO$_4$, pH 8.6 in a 0.2 mm Quartz-Suprasil cuvette (Hellma, GmbH & Co) and absorbance between 190 and 250 nm measured using a J-810

spectropolarimeter. Data was normalised by subtraction of the buffer spectrum from the sample spectrum before conversion into standard units of Δε (M$^{-1}$ cm$^{-1}$).

## Isothermal titration calorimetry

Isothermal titration calorimetry (ITC) to measure the interaction between TslA and TilA was performed in a buffer of 20 mM HEPES pH 7.5, 150 mM NaCl at 25 °C using the Microcal PEAQ-ITC system (Malvern Panalytical). The sample cell contained a 300 µl volume of 30 µM TslA. TilA at a starting concentration of 300 µM was present in the injection syringe. There was an initial injection of 0.4 µl of TilA, followed by 18 further injections of 2 µl with 60 s spacing between, with stirring at 750 rpm. Results were analysed using the Microcal PEAQ-ITC analysis software.

## Cryo-EM sample preparation and data acquisition

Four microliters of purified TilA-TslA-TlaA1-TlaA2 complex at 0.5 mg ml$^{-1}$ in Buffer B (Supplementary Table 5) was applied onto glow-discharged (60 s, 30 mA) 300 mesh R1.2/1.3 Au grids (Quantifoil). Grids were blotted for 3 s with blot force of +15 at 10 °C and 100% humidity, and plunge frozen in liquid ethane using a Vitrobot Mark IV (Thermo Fisher Scientific).

Data were collected in counted mode in EER format on a CFEG-equipped Titan Krios G4 (Thermo Fisher Scientific) operating at 300 kV with a Selectris X imaging filter (Thermo Fisher Scientific) with slit width of 10 eV and Falcon 4 direct detection camera (Thermo Fisher Scientific) at ×165,000 magnification, with a physical pixel size of 0.693 Å. Movies were recorded at a dose rate of 12.8 e$^-$/Å$^2$/s and 3.98 s exposure for a total dose of 50.9 e$^-$/Å$^2$.

## Cryo-EM data processing and map fitting

Patched motion correction, CTF parameter estimation, particle picking, extraction, and initial 2D classification were performed in SIMPLE 3.0[62]. All downstream processing was carried out in cryoSPARC v3.3.1[63], including global resolution estimation by Gold-standard Fourier shell correlation (FSCs) using the 0.143 criterion and local resolution estimation using an FSC threshold of 0.5.

The workflow for cryo-EM image processing is shown in Supplementary Fig. 11. Briefly, a total of 1,448 movies were collected and 1,309,475 particles extracted from motion-corrected micrographs then subjected to reference-free 2D classification in SIMPLE (k = 300) followed by two additional rounds of 2D classification in cryoSPARC (k = 120) using a 120 Å diameter spherical mask. Based on the resulting 2D class averages it was evident that particles suffered from preferred orientations. Particles were therefore selected from classes corresponding to the most divergent views but also containing strong secondary structure features. These 2D-cleaned particles (21,177) were subjected to multi-class ab initio reconstructions ($k = 2$) generating a poorly occupied (8044 particles, 38% of total particles) volume lacking the TslA C-terminal domain and another more occupied volume (13,133 particles, 62% of total). Particles belonging to the latter volume were selected for non-uniform refinement against their corresponding 30 Å lowpass-filtered volume, yielding a 7.3 Å map in which helical structure was evident. Further extensive classification and particle rebalancing in 2D and 3D space did not significantly improve map quality, indicating the sample likely also suffered from significant conformational heterogeneity.

The TilA-TslA-TlaA1-TlaA2 AlphaFold model was rigid body fit into the cryo-EM volume using the fitmap function in ChimeraX[64]. TlaA1 was emitted from the model for clarity.

## Biochemical assays for lipase activity

Lipase activity was determined by assessing the hydrolysis of Tween 20 measured through CaCl$_2$-mediated precipitation of the released fatty acid. To 100 µl aliquots of 2% Tween 20 in 20 mM HEPES pH 7.5,

150 mM NaCl, 5 mM CaCl$_2$, the protein of interest was added to a final concentration of 5 µM. Assays were performed in Greiner CELLSTAR® 96 well plates at 30°C. Plates were agitated for 30 s before measuring absorbance increase at 500 nm, using a TECAN infinite nano M+ reader. A buffer only sample was used as a negative control.

Phospholipase A$_1$ activity was assayed using the EnzChek™ Phospholipase A$_1$ Assay Kit (Thermofisher), according to the manufacturer's protocol. Briefly, purified protein at the concentrations indicated in the respective figure legends were diluted in the kit buffer, unless otherwise stated in the figure legends. Lipid substrate mixtures were added at a 1:1 ratio (v/v) to the protein solution. Assays were performed in a Greiner CELLSTAR® black 96-well plate. Fluorescence emission at 515 nm was measured following excitation at 470 nm at 30 °C using a TECAN infinite nano M+ reader. To test for PLA$_2$ activity, 2 mM PED6 was used in place of PED-A1, with all other components the same. For the Phospholipase A$_1$ assay, Lecitase® Ultra was used as the positive control. For the Phospholipase A2 assay, PLA2 from honeybee venom was used as the positive control. For all assays, the negative control consisted of the respective buffer with liposome mix in the absence of protein.

### Lipidomics

*S. aureus* strains USA300 and USA300 Δ*til1* harbouring pRAB11 or pRAB11_TslA-TlaA1-TlaA2 were grown overnight and subcultured to a final OD$_{600}$ 0.1 into fresh TSB medium containing Cml and supplemented with 5 mM CaCl$_2$. Cultures were grown at 37 °C with shaking for 2 h after which they were supplemented with 500 ng ml$^{-1}$ ATc. At 2 h and 6 h post supplementation, samples were withdrawn equivalent to 1 ml of OD$_{600}$ = 1. Cells were pelleted by centrifugation (800 $g$, 10 min), resuspended in 100 µL 1 X PBS and transferred to a glass tube and snap frozen. Total lipids from cells were extracted using the method of Bligh and Dyer[65]. Briefly, 375 µL of 2:1 (v/v) MeOH:CHCl$_3$ was added to the glass tube containing the frozen cell pellet and vortexed. The sample was agitated vigorously for a further 10-15 min, after which the sample was made biphasic by the addition of 125 µL of CHCl$_3$. The sample was then vortexed and 125 µL of H$_2$O was added. The solution was vortexed again before centrifugation at 1000 $g$ at RT for 5 min. The lower phase was transferred to a new glass vial and dried under nitrogen and stored at 4 °C until analysis.

Organic phases were suspended in 2:1 (v/v) MeOH:CHCl$_3$ and high-resolution mass spectrometry data were acquired by electrospray ionisation techniques using a Thermo Scientific™ Exactive™ Orbitrap mass spectrometer. Phospholipid species annotations were determined in reference to previous assignments[66] and the LIPID MAPS database (https://www.lipidmaps.org).

### Murine skin and soft tissue infection model

The murine infection model was performed as described previously[67,68] with minor modifications. Briefly, female C57BL/6JRccHsd mice (6 weeks) were obtained from Envigo. Female mice were used for these experiments to allow comparison with existing datasets using this infection model. Animals were housed in IVC cages. A maximum of five mice were kept in "Type 2 long" cages with wooden tubes as shelters along with a wooden chewing block and cellulose as cage enrichment. Feed and water were provided *ad libitum*. The facility was held at 22 +/- 1 °C, 50% relative humidity and day/night cycles of 12 h.

To induce abscess formation, bacteria were mixed with equal volumes of dextran beads (Cytodex-1 microcarriers; Sigma) prepared according to the manufacturer's instructions. For each strain, 200 µl volumes containing 5 ×10$^4$ colony forming units (c.f.u.) were injected subcutaneously into the flanks (left/right) of the mice. The allocation of mice to groups infected with different strains was random. After 48 h, mice were euthanized, each abscess was excised, homogenised in 1 ml 1 X PBS (Gibco from Life Technologies) and c.f.u. were enumerated on TSA (Becton Dickinson GmbH) plates. The two abscesses of each mouse were regarded as independent experiments. A One-way ANOVA was performed to determine statistical significance.

### Bioinformatic analysis

To identify homologues of TslA, blastP was used against the RefSeq database[69]. Accession lists generated from these searches were submitted to the webFlaGs server to identify genetic neighbourhoods[70] which were visualised using Clinker[71]. ProgressiveMauve was used to align whole genome shotgun sequencing results to the reference genome and to assess genetic differences in strains[72].

Proteins with structural homology to TslA were identified using Phyre2[29]. AlphaFold models of TslA and TelC were obtained from the AlphaFold Protein Structure Database[73,74]. AlphaFold Colab was used to model TslA in complex with TilA, TlaA1 and TlaA2[73,75]. Structural models were aligned using the cealign function in PyMOL V2.1.0[76] and rendered in Chimera X V1.4[64]. Protein alignments were carried out using MUSCLE v3.8.1551[77] and visualised with boxshade (https://github.com/mdbaron42/pyBoxshade).

### Statistics and reproducibility

No statistical method was used to predetermine the sample size. For the murine infection model, the sample size was in line with our ethical allowance (IMIT01/20 G), binding us to the use of 5 mice per group with two abscesses per mouse. The allocation of mice to groups was randomised, but for other experiments, the experimental design does not require randomisation. For statistical analysis at least three biological replicates were used. The statistical tests used as well as the number of replicates for each experiment are indicated in the respective figure legends. No data were excluded from the analyses. The Investigators were not blinded to allocation during experiments and outcome assessment.

### Reporting summary

Further information on research design is available in the Nature Portfolio Reporting Summary linked to this article.

## Data availability

Other data generated or analysed in this study are available within the article and its supplementary materials. Source data are provided with this paper with the exception of the microscopy images which are available at FigShare (https://doi.org/10.6084/m9.figshare.24717864.v1). Databases used are Alphafold (https://alphafold.ebi.ac.uk/), Refseq (https://www.ncbi.nlm.nih.gov/refseq/) and Lipid maps (https://www.lipidmaps.org). Source data are provided with this paper.

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

## Acknowledgements

This study was supported by the Wellcome Trust (through Investigator Awards 10183/Z/15/Z and 224151/Z/21/Z to TP), the Intramural Research Program of the NIH, NCI, Center for Cancer Research (awarded to SML), the German Centre of Infection Research (DZIF) to SH (TTU 08.708). NM holds a Walter Benjamin Fellowship (M2871/1-1), funded by the DFG (German Research Foundation). Additionally, we acknowledge infrastructural funding by the DFG in the frame of Germany´s Excellence Strategy—EXC 2124—390838134 (SH). SG is funded by the Newcastle-Liverpool-Durham BBSRC DTP2 Training Grant, project reference number BB/M011186/1 and YY by the China Scholarship Council. We would like to thank Dr James Grimshaw from the Newcastle University Image Analysis Unit for his advice and guidance in the analysis of our microscopy data. We thank Dr Helen Waller for her help in performing circular dichroism.

## Author contributions

S.G., N.M., S.H., T.K.S., S.M.L. and T.P. designed experiments. S.G., N.M., J.D., A.B., Y.Y., F.R.U., D.K. and T.K.S carried out experimental work. S.G., N.M. J.D., S.H., T.K.S. and T.P. undertook data analysis. T.P. wrote the manuscript and S.G. and T.P. edited the manuscript. All authors have approved the final version.

## Competing interests

The authors declare no competing interests.
