## [Peer Review File · Nature Communications]

REVIEWER COMMENTS

Reviewer #1 (Remarks to the Author):

Garrett and colleagues identify and characterize a new lipase enzyme secreted by the T7SS of *S. aureus*. Compared to known T7SS secreted proteins, this lipase (named TslA) has a unique backwards domain arrangement with the signal sequence at the C-terminus and lipase domain at the N-terminus. Despite this unusual arrangement, this protein is exported by the T7SS in a manner similar to 'normal' T7SS proteins. Additionally, the predicted lipase activity is confirmed biochemically and endows TslA with bactericidal activity. Overall, this work is both exciting and technically sound. I have only minor comments for the authors below.

Minor:

- line 52: the T7SSa and T7SSb nomenclature is new to me and I couldn't find any evidence of it being used in the reference cited at the end of this sentence. Perhaps the authors meant to cite a different article?
- Fig 1c: in the absence of locus tags, these gene clusters will not easily be found/analyzed by others. Please add additional information to figure and/or figure legend.
- Lines 115-150: Given that many of the cited papers for *S. aureus* T7SS use western blots as a means to detect protein secretion, the inclusion of a rationale statement for why the nanoluciferase assay was used in this study would be helpful.
- Lines 115-150: The use of locus tags throughout this paragraph makes it quite difficult to follow for someone not familiar with the genes. I would encourage the authors to consider naming these genes at the outset of the results section so that the rest of this section is more digestible to non-specialists.
- Line 153: I'm confused why the authors used Phyre2 analysis when AlphaFold used alongside FoldSeek has been shown to be much more accurate. Especially when the authors then go on to use alphafold for other analyses later in the paper.
- Line 270: I'm unfamiliar with the antibiotic melittin. It would be helpful to better explain what this molecule is and why it is a good control for this assay.
- Line 280: The lipidomics data are critical for the analysis of TslA function, so I'm not sure why they are 'buried in the supplement'. I recommend the authors bring these data into the main body of the paper.
- Line 299: "A strain lacking all *til1* genes was also less virulent than the wild type." Please add a second sentence explaining your interpretation of this result.

Reviewer #2 (Remarks to the Author):

This is an exciting and well written manuscript that identifies a novel T7SS substrate in *S. aureus*, which is a reverse-arranged toxin, TslA. Herein, the authors show that TslA that functions as a lipase and exhibits anti-staphylococcal activity upon secretion. The authors further investigate the requirement of TlaA chaperones for the secretion and toxicity of TslA. The findings are interesting and important for the field. Concerns exist that convincing evidence is not provided for the interaction of TlaA1 with TlsA/TlaA2 to substantiate claims made. Further, a role for TslA in virulence cannot be ruled out by the

data provided. Finally, to validate the role of the immunity factor TlIA, a knockout of 18 genes is used (including deletion of non-tlIA homologs like tsIB). The experiments are well executed and controlled, and in general support the claims made. I have some suggested experiments that would strengthen the study as well as some questions and minor comments below.

1. I don't see convincing evidence that TlA1 interacts with TlA2 or TslA (Fig 2c, Suppl Fig 4, Fig 4b). The only evidence is that they are found in the same fraction when all are overexpressed. However, there are no negative controls provided for the SDS-PAGE analysis regarding the specificity of the chaperone bands observed in peaks a and b (Fig 2e and 4a). Co-IP experiments would provide evidence for the complex/direct interaction of TslA-TsaA1-TsaA-2-TlIA. The statement in line 178-180 should be adjusted to reflect this.

2. The authors and other groups have previously shown that other T7SS toxins can promote virulence/immune modulation. So it seems that potential virulence due to TslA (and LXG proteins in general) is downplayed. The tsIA mutant in the Mouse model of skin abscess in figure 6 is not significant but is likely trending. The authors could comment on why the til1 deletion strain would have a phenotype, but not the tsIA mutant—for example, is this expected to be due to lack of tsIB or other non-tlIA genes? If USA300 does indeed encode multiple Tsl paralogs, individual and combination deletion strains would help assess the contribution of each to virulence. It would be helpful to provide P values, instead of “ns”. The n is variable across groups and is not provided in the figure legend so it is unclear how many replicates there are. Further, bacterial burden is not the only metric for virulence—even if bacterial burden is similar there may be differences in survival, inflammation, toxicity, etc. The conclusion on line 302 that “the major role of TslA is in interbacterial competition” is overstated and not directly supported by data since toxicity is only shown in *S. aureus* upon overexpression of the toxin in the absence of immunity factor genes. This claim should be softened.

3. A USA300 mutant deleted for 18 “tlIA-related genes” is used in Figures 5-6, but only 11 tlIA homologs are shown in Suppl Fig 6 so it is unclear what the other deleted genes are? The authors indicate that tsIB is deleted and that tsIA should not be expressed in this strain due to the lack of a start codon, but please further clarify exactly which genes are deleted in this strain. Deletion of non-relevant genes from the *lpl* loci lack of tsIA/B heavily convolutes the conclusions drawn for this strain in Figures 5-6. It would be better to use a clean deletion of the TslA-specific immunity factor, 00405.

4. Other LXG proteins seem to require interaction with chaperones for stability. Is this the case for TslA? Does deletion of TlA1 and TlA2 impact TslA secretion? Are the chaperones secreted in the absence of TslA?

5. Overall, the comparisons between TslA, 02786, and TslB and TslC are hard to follow in Lines 104-113, 150. It is unclear if these additional two copies of the four-gene locus are found within the same *S. aureus* strain since Fig1b shows diagrams in other strain backgrounds. Does ST398 encode both TslA and B? Or does any given *S. aureus* strain encode all three or just one full-length Tsl (and possibly an additional orphaned/truncated 02786 homolog)?

6. In Fig 1, accession NC_007795 is indicated as RN6390 but on Genbank this is strain NCTC8325. Please

confirm all accession numbers in these figures are correct.

7. An alignment is provided in Suppl Fig 2b, but what is the % identity across these paralogues? Are 02786 or TslB/C secreted/ exhibit lipase activity as well? Is 02786 encoded for near the T7SS locus?

8. A domain diagram could be helpful in Fig 1 to show the length of the LXG domain and N-terminal toxin domain.

9. In line 136-141, Suppl Fig 3a: it is hard to say if the slight increase in luminescence for tagged TlaA1 in WT vs *essC* mutant *S. aureus* is biologically meaningful. It would be nice to confirm *EssC*-dependent secretion of TlaA1 by Western blot.

10. In SFig 4, it appears that the TslA C-terminus may interact with itself. Can the authors address this? Is there any indication that TslA oligomerizes?

11. Are all TilA homologs shown in Suppl Fig 6 protective against TslA, or just 00405? Are different immunity factors required for TslB/C?

12. Does TilA also inhibit lipase "A2" activity? (Fig 3d)

13. Has the predicted complexes from Fig 2e or Fig 4 been assessed under native/non-denaturing conditions to determine their size/stoichiometry?

14. Is TilA secreted? Line 333 indicates that TilA is extracellular, but I don't think this was shown. TslA is compared to TelCl in *S. intermedius*. Is TilA expected to have a similar mechanism of neutralization to TipC?

15. Figure 5: Inhibition of *S. aureus* growth is observed upon overexpression of TslA-TlaA1-TlaA2. Has CFU been analyzed to determine if it is bactericidal? Is there any indication that WT *S. aureus* may natively compete with either *S. aureus* lacking the immunity factor or other bacteria? This could be tested easily in in vitro predator prey assays.

16. A control should be added to Figure 5a, showing that toxicity is not induced by expression of TlaA1 & 2 alone.

17. In Figure 5b, the catalytic residue mutants exhibit reduced toxicity. The expression of these mutants was validated in Suppl Fig 8, but their secretion was not confirmed. This control should be performed to confirm the lack of toxicity is truly due to inactivity of the lipase, not due to lack of secretion.

18. In Figure 5d, the authors state that Δ til1 experiences TslA dependent membrane depolarization/instability using these fluorescent dyes but the representative images do not support these claims.

a. the positive control for USA300 WT is weak and fairly similar to the negative control.

b. The positive vs negative controls are better for the Δ til1 strain but there is hardly any Sytox green

staining in the TslA expressing condition and similar DISC staining to WT USA300.

c. Line 273-276 indicates that the cells that became depolarized also stained for Sytox Green, but looking at the images this is not clear.

19. Figure 5e: The figure legend says that quantitative analysis of the microscopy isn't possible, so the cells were binned into high/low categories, but this graph has a numerical axis. Not sure I understand where these numbers came from and why stats couldn't be performed.

Minor

It would be helpful to highlight the GxL motif in Supp Fig 2b.

Typically, one of the Laps contain an DUF3130 domain. Is this the case for TlaA1 or TlaA2?

Throughout it could be helpful to label graphs in figures. For example, supernatant vs cytosolic graphs in Supp Fig 3 a-b.

The Laps are named twice in Lines 146-149 (could be condensed), but the abbreviation "TlaA" is not defined. The Lap names in line 147 are missing the uppercase, "A".

It would be helpful to define/describe LPL loci more in the text (line 150, Fig 1 and Supp Fig 6).

Fig 2c chart is blurry.

Figure 3 and 5 graphs are small, and it is difficult to tell colors and symbols. Making figure larger and using a more distinct color scheme would be helpful.

Line 188: should be "Supplementary Fig. 5e,f"

Lines 212-213 say that three of four loci in Suppl Fig6 encode a TslA homolog, but only two loci in this figure depict tslA (red arrow).

No positive control is included for Fig 3b?

Can the authors discuss why the D244A mutation abrogated lipase activity in Suppl Fig 5g but not fully in Figure 3a-b?

Statistical test information is not provided for figure 1e. Please indicate the post-hoc test used for multiple comparisons in the two-way ANOVAs in Fig 1e and supplemental Fig 3 and for the one-way ANOVA in Fig 6. In line 717, the definition of significant p value is duplicated.

Lines 271 and 273: I don't think figure 5c is relevant in these lines, is it supposed to just be figure 5d referenced?

Lines 311-314: a reverse GxW motif was previously identified in Mtb T7SS substrate EsxE and shown to be important for function.

Reviewer #3 (Remarks to the Author):

Summary and key findings:

This is a review of an excellent manuscript by Stephen R. Garrett et al. titled “An interbacterial lipase toxin with an unprecedented reverse domain arrangement defines a new class of type VII secretion system effector” characterizing a novel type of effector translocated by the specialized T7SS of *S. aureus*. Authors find that TslA secretion requires chaperone factors Tla1 and Tla2 which interact with the C-terminal LXG domain of TslA. Further, authors characterize a phospholipase activity of TslA and demonstrate its ability to disrupt staphylococcal membranes when not inhibited by TilA immunity factor. TslA does not contribute to *S. aureus* USA300 virulence in the mouse skin abscess model of infection, therefore authors suggest it functions primarily in the bacterial antagonism.

Validity:

Based on the experimental data presented authors make valid conclusions as to the function of the TslA toxin and its interactions with Tla1, Tla2 and TilA. While the cryo-EM data is not conclusive and is missing density within potentially flexible regions, overall, it falls in line with the rest of the experimental evidence presented.

Significance:

This work is of interest to the field of *S. aureus* protein secretion and makes a significant contribution to our understanding of T7SS and unique toxins translocated by this specialized system. Since LXG domains are usually found at the N-terminus of T7SS effectors, characterization of TslA's unique reverse C-terminal orientation of LXG-like domain, not only uncovers a unique effector type, but also demonstrates plasticity of T7SS to export proteins with a reverse domain arrangement.

Methodology and analytical approach:

Experimental approaches are appropriate for this study. Data is of good quality and the manuscript is well written. Suitable analytical approaches were applied to assess protein-protein interactions and lipase activity. Genetic experiments were appropriately designed and included controls.

Clarity and context

Background and broader context are well presented and are clear to the scope of the study.

Suggestions:

It would be useful if authors included a local resolution map to supplement the average 7.3Å resolution map in Fig. 4b. This is important to evaluate the overall quality of the map. Workflow diagram for refinements could be useful as well.

If DUF576-family lipoproteins are prevalent and shed from the *S. aureus* membrane (causing potent

immune stimulation) which would inhibit extracellular TslA, how do authors envision TslA exerting its antibacterial activity.

Perhaps the statement in line 302 that TslA is required for bacterial antagonism is too strong since investigation of the physiological role of this toxin has not been exhausted, particularly it has not been explored using bacterial competition assays or other disease models.

Reviewer #4 (Remarks to the Author):

Reviewer #1 (Remarks to the Author):

Garrett and colleagues identify and characterize a new lipase enzyme secreted by the T7SS of *S. aureus*. Compared to known T7SS secreted proteins, this lipase (named TslA) has a unique backwards domain arrangement with the signal sequence at the C-terminus and lipase domain at the N-terminus. Despite this unusual arrangement, this protein is exported by the T7SS in a manner similar to 'normal' T7SS proteins. Additionally, the predicted lipase activity is confirmed biochemically and endows TslA with bactericidal activity. Overall, this work is both exciting and technically sound. I have only minor comments for the authors below.

We thank the reviewer for their positive comments.

Minor:

-line 52: the T7SSa and T7SSb nomenclature is new to me and I couldn't find any evidence of it being used in the reference cited at the end of this sentence. Perhaps the authors meant to cite a different article?

This is the correct citation - in Abdallah *et al.*, 2007, the Type VIIb nomenclature is introduced in Figure 5.

-Fig 1c: in the absence of locus tags, these gene clusters will not easily be found/analyzed by others. Please add additional information to figure and/or figure legend.

We have now added the locus tags to these gene clusters. This is now Fig 1b in the revised manuscript.

-Lines 115-150: Given that many of the cited papers for *S. aureus* T7SS use western blots as a means to detect protein secretion, the inclusion of a rationale statement for why the

nanoluciferase assay was used in this study would be helpful.

Whilst immunoblotting has been used in the past for assaying T7SS substrate secretion, analysis is not quantitative. Moreover, due to poor levels of secretion in laboratory media, supernatant samples must be concentrated (often by TCA precipitation), which introduces further variability into the analysis. By contrast, the nanoluciferase assay provides reproducible, quantifiable data with greater sensitivity (without the need for sample precipitation) and so is a much more accurate measure of the secretion of T7SS substrates than western blotting. We have now added a sentence to the manuscript to explain the rationale for using this method (line 125):

'This approach is more robust than western blotting which has been used previously to assess secretion because it is quantitative and avoids the requirement to concentrate supernatant proteins by precipitation²⁸'

-Lines 115-150: The use of locus tags throughout this paragraph makes it quite difficult to follow for someone not familiar with the genes. I would encourage the authors to consider naming these genes at the outset of the results section so that the rest of this section is more digestible to non-specialists.

We thank the reviewer for this comment and agree that as written it was quite clunky. We have taken the reviewer's advice and have now renamed the proteins at the outset (line 96 onwards). We hope this will make it more digestible to the general reader.

-Line 153: I'm confused why the authors used Phyre2 analysis when AlphaFold used alongside FoldSeek has been shown to be much more accurate. Especially when the authors then go on to use AlphaFold for other analyses later in the paper.

We have repeated this analysis using FoldSeek which has given similar results, with hits against several DUF 4176 lipases. At the outset of our work these tools were not available which is why we used Phyre2. As both methods of analysis give similar results we have not updated this in text.

-Line 270: I'm unfamiliar with the antibiotic melittin. It would be helpful to better explain what this molecule is and why it is a good control for this assay.

We have added the following text at line 286 to explain the rationale for using melittin:

'Melittin, an amphipathic helical peptide from honeybee venom, was used as a positive control as it forms toroidal pores in membranes of bacteria including *S. aureus*^{43,44}.'

-Line 280: The lipidomics data are critical for the analysis of TslA function, so I'm not sure why they are 'buried in the supplement'. I recommend the authors bring these data into the main body of the paper.

We have moved the lipidomics data as suggested. It is now Figure 6 of the revised manuscript.

-Line 299: "A strain lacking all *til1* genes was also less virulent than the wild type." Please add a second sentence explaining your interpretation of this result.

Strains which have previously had these lipoprotein genes deleted have shown a reduced virulence phenotype and we have clarified this and cited the relevant literature. We have added the following text (starting at line 314) to account for the reduced virulence when these genes are absent:

'The Til1 lipoproteins are known to be shed from the surface of wild type USA300 where they promote immune stimulation, which presumably accounts for the reduced virulence seen when the encoding genes are deleted^{36,37}. We therefore investigated whether TslA contributed to virulence of *S. aureus* in the skin abscess model. As shown in Fig. 7, inactivation of the T7SS by deletion of *essC* resulted in a strong reduction in bacterial burden compared with the wild type USA300 strain. A strain lacking all *til1* genes was also significantly less virulent than the wild type, in agreement with prior observations³⁷.'

Reviewer #2 (Remarks to the Author):

This is an exciting and well written manuscript that identifies a novel T7SS substrate in *S. aureus*, which is a reverse-arranged toxin, TslA. Herein, the authors show that TslA that functions as a lipase and exhibits anti-staphylococcal activity upon secretion. The authors further investigate the requirement of TlaA chaperones for the secretion and toxicity of TslA. The findings are interesting and important for the field. Concerns exist that convincing evidence is not provided for the interaction of TlaA1 with TlsA/TlaA2 to substantiate claims

made. Further, a role for TslA in virulence cannot be ruled out by the data provided. Finally, to validate the role of the immunity factor TilA, a knockout of 18 genes is used (including deletion of non-tilA homologs like tsIB). The experiments are well executed and controlled, and in general support the claims made. I have some suggested experiments that would strengthen the study as well as some questions and minor comments below.

We thank the reviewer for their positive comments and for their suggestions to strengthen our study, which we address in the specific responses below.

1. I don't see convincing evidence that TlaA1 interacts with TlaA2 or TslA (Fig 2c, Suppl Fig 4, Fig 4b). The only evidence is that they are found in the same fraction when all are overexpressed. However, there are no negative controls provided for the SDS-PAGE analysis regarding the specificity of the chaperone bands observed in peaks a and b (Fig 2e and 4a). Co-IP experiments would provide evidence for the complex/direct interaction of TslA-TsaA1-TsaA-2-TilA. The statement in line 178-180 should be adjusted to reflect this.

We have now repeated the purification of TslA_{CT}-TlaA1-TlaA2. This time we first used Ni-affinity chromatography to purify his-tagged TslA_{CT}, then taking the peak fractions from this we further purified using streptactin resin pulling on the Strep tag of TlaA2. This was then followed by SEC, the results of which are presented in new figure parts for Fig 2d and Fig 2e. The TlaA1 protein in this construct is supplied with a Myc tag. Fig 2f shows the result of western blotting for His (TslA), Strep (TlaA1) and Myc (TlaA2) (uncropped blots are also included in Supplementary Fig. S10). Given the strict purification regime we have used here we think we are justified in concluding that all three proteins form a complex. However, we acknowledge that we cannot conclude that TlaA2 interacts directly with TslA_{CT} (rather than via interactions TlaA1), and we have refrained from implying direct interactions. We have adjusted the title of this section (line 165) to read

'TlaA1 and TlaA2 form a complex with the C-terminal 'LXG-like' domain of TslA'.

And have clarified the text as requested (line 192 in the revised manuscript) to read

'We next co-overproduced TlaA1 with a Strep tag and Myc-tagged TlaA2 alongside TslA_{CT} that had been provided with a His tag. Double affinity purification first with HisTrap and then with StrepTrap columns, followed by size exclusion chromatography resulted in isolation of a complex of all three proteins (Fig. 2d,e). We conclude that TlaA1 and TlaA2 form a tripartite complex with the C-terminal domain of TslA and that these three proteins are likely co-secreted as a complex by the T7SS.'

2. The authors and other groups have previously shown that other T7SS toxins can promote virulence/immune modulation. So it seems that potential virulence due to TslA (and LXG proteins in general) is downplayed. The tsIA mutant in the Mouse model of skin abscess in figure 6 is not significant but is likely trending. The authors could comment on why the til1 deletion strain would have a phenotype, but not the tsIA mutant—for example, is this expected to be due to lack of tsIB or other non-tilA genes? If USA300 does indeed encode multiple Tsl paralogs, individual and combination deletion strains would help assess the contribution of each to virulence. It would be helpful to provide P values, instead of "ns". The n is variable across groups and is not provided in the figure legend so it is unclear how many replicates there are. Further, bacterial burden is not the only metric for virulence—even if bacterial burden is similar there may be differences in survival, inflammation, toxicity, etc.

The conclusion on line 302 that "the major role of TslA is in interbacterial competition" is overstated and not directly supported by data since toxicity is only shown in *S. aureus* upon overexpression of the toxin in the absence of immunity factor genes. This claim should be softened.

We have now modified the text to reflect the points raised here and by the other reviewers. We have acknowledged that there is a trend to reduced virulence for the tsIA mutant strain

that is not statistically significant (line 322) and added all the p-values into the legend for this figure. USA300 only has one full length Tsl1 protein, TslA. It lacks TslC, and TslB carries a point mutation which ablates the targeting domain. We have acknowledged that there might be a cumulative effect on virulence of more than one toxin. The phenotype of the *til1* deletion strain is now also expanded on in response to a comment from reviewer #1. The revised text (lines 311 - 325) now reads:

'Previous studies have reported a role for the *S. aureus* T7SS in murine abscess models of infection⁴⁵⁻⁴⁸. A strain lacking the *til1* cluster along with the *tsIA* cassette encoded on the ν Sa α island of USA300 also led to a significant reduction in bacterial burden in a mouse skin-infection model³⁷. The Til1 lipoproteins are known to be shed from the surface of wild type USA300 where they promote immune stimulation, which presumably accounts for the reduced virulence seen when the encoding genes are deleted^{36,37}. We therefore investigated whether TslA contributed to virulence of *S. aureus* in the skin abscess model. As shown in Fig. 7, inactivation of the T7SS by deletion of *essC* resulted in a strong reduction in bacterial burden compared with the wild type USA300 strain. A strain lacking all *til1* genes was also significantly less virulent than the wild type, in agreement with prior observations³⁷. However, no statistically significant difference in bacterial burden was observed between USA300 and USA300 Δ *tsIA*, although there was some trend towards decreased virulence of the *tsIA* mutant in this infection model. (Fig. 7). It is possible that the virulence defect observed with the *essC* mutant is a cumulative effect resulting from the lack of secretion of TslA alongside other T7SS toxins.'

3. A USA300 mutant deleted for 18 "tilA-related genes" is used in Figures 5-6, but only 11 tilA homologs are shown in Suppl Fig 6 so it is unclear what the other deleted genes are? The authors indicate that *tslB* is deleted and that *tslA* should not be expressed in this strain due to the lack of a start codon, but please further clarify exactly which genes are deleted in this strain. Deletion of non-relevant genes from the *lpl* loci lack of *tslA/B* heavily convolutes the conclusions drawn for this strain in Figures 5-6. It would be better to use a clean deletion of the TslA-specific immunity factor, 00405.

We apologise for not being clearer here. The locus depicted in Supplementary Fig. 6 (Supplementary Fig. 8 in the revised manuscript) shows the 4 LPL loci of NCTC8325, however, additional non-identical copies of *til1* genes are found at the LPL0 locus in USA300. We have now clarified this by showing the loci in USA300 as Supplementary Fig. 7b, and on this figure we have also included the boundaries of the deletions. There are no additional genes deleted in this strain other than *til1*, *tsl1* and *tla* genes.

We cannot rule out that other Til1 proteins from USA300 may also interact with TslA. Since there are a further 17 in addition to TilA it would require a very large amount of additional work to determine which of these we might also need to delete (and in which combinations) to ensure that we would have a strain that lacked any TslA immunity. This is why we opted to use a strain lacking any proteins of this family.

4. Other LXG proteins seem to require interaction with chaperones for stability. Is this the case for TslA? Does deletion of TlaA1 and TlaA2 impact TslA secretion? Are the chaperones secreted in the absence of TslA?

Full length TslA can be stably produced in *E. coli* without the two small partner proteins (Supplementary Fig. 6), this contrasts with our prior work on EsaD which is very unstable when produced in *E. coli* if its partners are lacking. A construct that only produces TslA but not the small partners is not secreted by the T7SS (Fig. 1, Supplementary Fig. 3, Supplementary Fig. 4). We have now performed an additional experiment to address the reviewer's question as to whether the small partners are secreted in the absence of TslA. Fig S3a and S4b shows that neither protein is secreted in the absence of TslA. We also mention this in the revised text (line 146 - 154)

'Previous studies have shown that some small Lap partner proteins are co-secreted with their cognate LXG toxin^{e.g.20} whereas others may be retained in the cytoplasm¹⁹. To determine whether TlaA1 or TlaA2 are also secreted by the T7SS, pep86 fusions were constructed to each and co-produced alongside the other small partner and TslA. The extracellular luminescent signal from either of these constructs was significantly greater in a strain where the T7SS was functional (Fig 1e; supplementary Fig. 3b, Supplementary Fig. 4a,b) although it was higher for the TlaA2 construct than for TlaA1. Furthermore, when *tslA* was absent from these constructs, neither TlaA protein was secreted by the T7SS, confirming that TslA, TlaA and TlaA2 are all co-secreted (Supplementary Fig. 3a,b, Supplementary Fig. 4).

5. Overall, the comparisons between TslA, 02786, and TslB and TslC are hard to follow in Lines 104-113, 150. It is unclear if these additional two copies of the four-gene locus are found within the same *S. aureus* strain since Fig1b shows diagrams in other strain backgrounds. Does ST398 encode both TslA and B? Or does any given *S. aureus* strain encode all three or just one full-length Tsl (and possibly an additional orphaned/truncated 02786 homolog)?

We apologise that this was not written clearly. We have now undertaken further analysis using all available *S. aureus* genome sequences from RefSeq to determine the percentage of strains encoding each of the three homologues and included this analysis as a new supplementary table (Supplementary Table 1). The text has been expanded to include this information (lines 107-122):

'We noted that the N-terminal region of TslA was polymorphic across different *S. aureus* strains (Supplementary Fig. 1). Up to two further copies of this four gene locus can be found encoded at other locations on the *S. aureus* chromosome in a strain-dependent manner (Fig. 1c). In commonly studied strains such as USA300, NCTC8325 (the parental strain of RN6390) and COL, only one further TslA homologue is encoded, SAOUHSC_02786 (TslB), the gene for which carries a frame shift at codon 345 and it therefore does not align with the final approximately 100 amino acids of TslA (Supplementary Fig. 2a). However, in other strains such as ST398 the frameshift is absent, and the protein aligns with TslA and with a third homologue, CO08_0212 (TslC), encoded by strain CO-08, along its entire length (Supplementary Fig. 2b). Across *S. aureus* strains, TslA proteins share 75-94% identity, with most of the sequence variability falling in the polymorphic N-terminal region. By contrast, in strain CO-08 which encodes full length variants of all three Tsl1 proteins, TslA shares 47% identity with TslB and 42% identity with TslC, while TslB shares 47% identity with TslC. Analysis of the *S. aureus* genome sequences present in the RefSeq database indicated that 78.5% encode a full length copy of TslA, 41.3% a full length copy of TslB and only 13.3% a full length copy of TslC (Supplementary Table 1).'

6. In Fig 1, accession NC_007795 is indicated as RN6390 but on Genbank this is strain NCTC8325. Please confirm all accession numbers in these figures are correct.

We apologise for this. It should indeed be NCTC8325 and we have fixed this in the figure and in the text. The accession numbering in the figures are correct.

7. An alignment is provided in Suppl Fig 2b, but what is the % identity across these paralogues? Are 02786 or TslB/C secreted/ exhibit lipase activity as well? Is 02786 encoded for near the T7SS locus?

We have now added percentage identities in the text as requested. See the response to point 5 above where we now include information about the percentage identities between the homologues.

Currently all our work has focused on TslA and we have not undertaken any experiments with TslB and TslC other than *in silico* analysis. 02786 (TslB) is not encoded near the T7SS locus. Although outside the scope of the present study, in future we plan to examine the distribution of these toxins and the specificity of the polymorphic lipase domains to provide further information about the range of T7SS targets.

8. A domain diagram could be helpful in Fig 1 to show the length of the LXG domain and N-terminal toxin domain.

Interpro and other domain prediction software are unable to identify any domain for the CT LXG-like domain of TslA, likely due to the reverse configuration of this domain. As such, we opted to provide a structural model generated by AlphaFold for TslA rather than include a domain diagram depicting the NT lipase domain alone.

9. In line 136-141, Suppl Fig 3a: it is hard to say if the slight increase in luminescence for tagged TlaA1 in WT vs *essC* mutant *S. aureus* is biologically meaningful. It would be nice to confirm *EssC*-dependent secretion of TlaA1 by Western blot.

We use the Nanoluciferase assay as it is much more sensitive than immunoblotting for the secretion of T7SS substrates and allows for quantitative analysis, which is not possible by Western blotting as supernatant samples have to be TCA precipitated which introduces high variability between samples. We have addressed our rationale for using this method in our response to reviewer #1

10. In SFig 4, it appears that the TslA C-terminus may interact with itself. Can the authors address this? Is there any indication that TslA oligomerizes?

We see no evidence of TslA oligomerisation when it is expressed and purified in the absence of partner proteins (e.g. SEC trace in Supplementary Fig S6).

11. Are all TilA homologs shown in Suppl Fig 6 protective against TslA, or just 00405? Are different immunity factors required for TslB/C?

At present we do not know which, if any, of the additional Til1 proteins provide protection against TslA activity/toxicity and unravelling this will be an interesting future direction of study, but is outside the scope of this present work.

12. Does TilA also inhibit lipase "A2" activity? (Fig 3d)

We have now repeated the fluorescent phospholipase assay using PED6 as a substrate to test for TilA inhibition and included this as new data (Supplementary Fig. 8b). The results show that TilA also inhibits this activity. It should be noted that to do these new experiments it was necessary to purify new batches of both TslA and TilA proteins so we cannot directly compare these results with those for PED-A1.

13. Has the predicted complexes from Fig 2e or Fig 4 been assessed under native/non-denaturing conditions to determine their size/stoichiometry?

The complexes have been subjected to size exclusion chromatography which is non-denaturing (Fig 2d, Fig 4a). It should be noted however that the Cryo-EM analysis and AlphaFold models indicate the complex comprises extended helices. Our studies with EsaD complexes (which have a similar extended conformation) have shown that they do not migrate at the expected size by size exclusion chromatography, calibrated using globular proteins (Yang, Boardman *et al.*, 2023). Moreover, the small partner proteins are <15Kda in size and the error on size determination is likely to be at least this, making it very difficult to determine stoichiometry.

14. Is TilA secreted? Line 333 indicates that TilA is extracellular, but I don't think this was shown. TslA is compared to TelCl in *S. intermedius*. Is TilA expected to have a similar mechanism of neutralization to TipC?

TilA and other Til1 proteins have been studied in the past and shown to be secreted lipoproteins (e.g. Ref³⁶). Based on the crystal structure of TipC, the binding of TilA to TslA is

likely to be similar, however, higher resolution structural data for the TslA-TilA complex would be needed to confirm this.

15. Figure 5: Inhibition of *S. aureus* growth is observed upon overexpression of TslA-TlaA1-TlaA2. Has CFU been analyzed to determine if it is bactericidal? Is there any indication that WT *S. aureus* may natively compete with either *S. aureus* lacking the immunity factor or other bacteria? This could be tested easily in in vitro predator prey assays.

We have not carried out CFU assays because the growth curve indicates that growth continues (albeit at a slower rate) after toxin induction. So CFU counts would not clarify whether the toxin is bactericidal or bacteriostatic. The situation is confounded because *S. aureus* has lipid repair pathways, and the bacteria will attempt to repair the TslA-induced damage which may be why we see continued growth. This is likely also why only a fraction of the cells is permeabilised when examined by microscopy. None-the-less, given the extensive permeabilisation in those cells stained with Sytox Green we think it unlikely that the cells could survive with such significant membrane damage.

We are still trying to optimise our predator/prey assays with TslA to examine which bacteria are susceptible to TslA-dependent killing and do not have any data in this regard. However, the TslA-dependent self-killing of the *S. aureus* immunity mutant is a good indication that the toxin acts on bacteria. We also discuss the presence of orphan immunity genes in other staphylococci implying that these species are likely targets of TslA toxicity (lines 375 – 3767).

16. A control should be added to Figure 5a, showing that toxicity is not induced by expression of TlaA1 & 2 alone.

Our results show that the active site point mutants of TslA are still secreted (point 17 below) but are no longer toxic. As we know that TlaA1 and TlaA2 are co-secreted with TslA, the fact that the TslA point mutants lack toxicity indicates that that TlaA1 and TlaA2 are not toxic.

17. In Figure 5b, the catalytic residue mutants exhibit reduced toxicity. The expression of these mutants was validated in Suppl Fig 8, but their secretion was not confirmed. This control should be performed to confirm the lack of toxicity is truly due to inactivity of the lipase, not due to lack of secretion.

We have now performed Nanoluc secretion assays with the active site mutants and demonstrated they are still secreted by the T7SS (Supplementary Fig S3a, Supplementary Fig 4b). Furthermore, we have demonstrated that TlaA1 and TlaA2 are not secreted by the T7SS in the absence of TslA (response to point 4 above), confirming the toxicity observed for the *til1* immunity mutant is due to the lipase activity of secreted TslA.

18. In Figure 5d, the authors state that $\Delta til1$ experiences TslA dependent membrane depolarization/instability using these fluorescent dyes but the representative images do not support these claims.

a. the positive control for USA300 WT is weak and fairly similar to the negative control.

b. The positive vs negative controls are better for the $\Delta til1$ strain but there is hardly any Sytox green staining in the TslA expressing condition and similar DISC staining to WT USA300.

c. Line 273-276 indicates that the cells that became depolarized also stained for Sytox Green, but looking at the images this is not clear.

We have now provided more representative images.

19. Figure 5e: The figure legend says that quantitative analysis of the microscopy isn't possible, so the cells were binned into high/low categories, but this graph has a numerical axis. Not sure I understand where these numbers came from and why stats couldn't be performed.

We apologise for not explaining this more clearly. Quantitative analysis could not be performed as the exposure for both DiSC and Sytox were set against the negative control, meaning when Sytox was measured for the positive control, some cells reached the maximum value, therefore likely exceeding this value and prohibiting true quantitative analysis. There are still two distinct groups of intact and permeabilised cells hence why cells were binned rather than analysed quantitatively. We have now clarified this in text (lines 486 – 488).

Minor:

It would be helpful to highlight the GxL motif in Supp Fig 2b.

We have now highlighted this on the figure.

Typically, one of the Laps contain an DUF3130 domain. Is this the case for TlaA1 or TlaA2?

No domains are predicted for either TlaA1 or TlaA2, using any of the conventional domain prediction programs (eg interpro, ncbi domain search, phyre2 etc.)

Throughout it could be helpful to label graphs in figures. For example, supernatant vs cytosolic graphs in Supp Fig 3 a-b

We have now presented the nanoluciferase data as relative values to simplify the data for the reader. All raw data has now been included in Supplementary Fig 4, where titles have been included for clarity.

The Laps are named twice in Lines 146-149 (could be condensed), but the abbreviation “TlaA” is not defined. The Lap names in line 147 are missing the uppercase, “A”.

We have now clarified the Tla abbreviation in the text (line 162; type VII secreted lipase accessory protein). The Lap names in line 147 relate to the general family – the lipase family in general is named Tsl1, and the Laps Tla1/Tla2 and the immunity Til1. So for example the partners of TslB will be TlaB1 and TlaB2, using the Tla1 and Tla2 nomenclature, and the immunity TilB.

It would be helpful to define/describe LPL loci more in the text (line 150, Fig 1 and Supp Fig 6).

These are discussed later in text in the section ‘TslA interacts tightly with an inhibitory protein, SAOUHSC_00405/TilA’

Fig 2c chart is blurry.

We have now re-labelled this figure to make it clearer, however, this figure was used at the dpi provided by AlphaFold so the chart itself cannot be made higher resolution.

Figure 3 and 5 graphs are small, and it is difficult to tell colors and symbols. Making figure larger and using a more distinct color scheme would be helpful.

We have now updated the graphs to make them more accessible to the reader.

Line 188: should be “Supplementary Fig. 5e,f”

This has been corrected.

Lines 212-213 say that three of four loci in Suppl Fig6 encode a TslA homolog, but only two loci in this figure depict tsIA (red arrow).

We have now updated the figure to include USA300 LPL loci. Neither USA300 or NCTC8325/RN6390 encode a homologue of TslC at the LPLI locus, however, Fig 1b shows an intact version of *tsIB* at the LPLIII locus of ST398 and a copy of *tsIC* at the LPLI locus of CO-08.

No positive control is included for Fig 3b?

Positive controls were provided with the PLA1 and PLA2 assays, however, PED6 (a PG derivative) had to be used to test for phospholipase A2 activity in place of BODIPY A2 (an aPC derivative), as aPC is not a substrate for TslA. The PLA2 positive control was not active against PED6 (likely due to it having specificity for aPC rather than PG lipids). The PLA1 positive control will have activity against PED6, however, the A2 chain of PED6 is labelled and the PLA1 positive control can only cleave the A1 chain therefore cannot be used as a control. Hence, for this experiment, we used the active site point mutants as a negative control to show relative lipase activity against a PLA2 substrate.

Can the authors discuss why the D244A mutation abrogated lipase activity in Suppl Fig 5g but not fully in Figure 3a-b?

These assays use different buffer conditions which likely accounts for the observed differences in activity seen. The role of the Asp residue in the widespread serine protease catalytic triad (SDH) is to stabilise the charge of the His allowing for nucleophilic attack of the substrate by the Ser. As such, the Asp is less essential than the other residues in this catalytic triad, with some enzymes having alternative residues to the Asp, and some consisting of the Ser-His alone (e.g. <https://doi.org/10.1110/ps.035436.108>).

Statistical test information is not provided for figure 1e. Please indicate the post-hoc test used for multiple comparisons in the two-way ANOVAs in Fig 1e and supplemental Fig 3 and for the one-way ANOVA in Fig 6. In line 717, the definition of significant p value is duplicated.

We have changed how we have presented the data in fig 1e, and Supplementary Fig 3. We have clarified how we have undertaken statistical analysis on line 436 (A Two-way ANOVA was performed, using multiple comparison between groups to determine statistical significance (n.s. $p > 0.05$; * $p < 0.05$; *** $p < 0.001$; **** $p < 0.0001$). This has also been updated in the legend for Fig 6 (now Fig 7 in the revised manuscript (Multiple comparison of each strain against the wild type was used to demonstrate deviation from the control (n.s. $p > 0.05$; * $p < 0.05$). The p-values are as follows for wild type v.: *til1* = 0.0392; *essC* = 0.0197; *tsIA* = 0.1622). The duplication has been deleted.

Lines 271 and 273: I don't think figure 5c is relevant in these lines, is it supposed to just be figure 5d referenced?

We have separated this into a new paragraph. The complementation with *TilA* is separate from the active site section and is clearer if it now sits in a separate paragraph.

Lines 311-314: a reverse GxW motif was previously identified in Mtb T7SS substrate EsxE and shown to be important for function.

We thank the reviewer for drawing our attention to this finding (<https://www.nature.com/articles/s41467-020-20533-1>). We have examined this manuscript carefully. We note that EsxE has a canonical WxG motif at the turn between two alpha helices and that in this study the authors show that the W in this motif is essential for secretion. By contrast the GxW motif is located at the C-terminal end of the EsxE rather than in the turn the W of this motif forms an oligomerisation interface with partner protein EsxF, rather than forming part of the secretion motif. Although this is interesting, we think it is not analogous to the GxL motif of TslA which we have shown contributes to the secretion of the protein.

Reviewer #3

Summary and key findings:

This is a review of an excellent manuscript by Stephen R. Garrett et al. titled “An interbacterial lipase toxin with an unprecedented reverse domain arrangement defines a new class of type VII secretion system effector” characterizing a novel type of effector translocated by the specialized T7SS of *S. aureus*. Authors find that TslA secretion requires chaperone factors Tla1 and Tla2 which interact with the C-terminal LXG domain of TslA. Further, authors characterize a phospholipase activity of TslA and demonstrate its ability to disrupt staphylococcal membranes when not inhibited by TilA immunity factor. TslA does not contribute to *S. aureus* USA300 virulence in the mouse skin abscess model of infection, therefore authors suggest it functions primarily in the bacterial antagonism.

Validity:

Based on the experimental data presented authors make valid conclusions as to the function of the TslA toxin and its interactions with Tla1, Tla2 and TilA. While the cryo-EM data is not conclusive and is missing density within potentially flexible regions, overall, it falls in line with the rest of the experimental evidence presented.

Significance:

This work is of interest to the field of *S. aureus* protein secretion and makes a significant contribution to our understanding of T7SS and unique toxins translocated by this specialized system. Since LXG domains are usually found at the N-terminus of T7SS effectors, characterization of TslA’s unique reverse C-terminal orientation of LXG-like domain, not only uncovers a unique effector type, but also demonstrates plasticity of T7SS to export proteins with a reverse domain arrangement.

Methodology and analytical approach:

Experimental approaches are appropriate for this study. Data is of good quality and the manuscript is well written. Suitable analytical approaches were applied to assess protein-protein interactions and lipase activity. Genetic experiments were appropriately designed and included controls.

Clarity and context

Background and broader context are well presented and are clear to the scope of the study.

We thank the reviewer for these positive comments.

Suggestions

It would be useful if authors included a local resolution map to supplement the average 7.3Å resolution map in Fig. 4b. This is important to evaluate the overall quality of the map. Workflow diagram for refinements could be useful as well.

We have now included this information as an additional supplementary figure – Supplementary Fig 11.

If DUF576-family lipoproteins are prevalent and shed from the *S. aureus* membrane (causing potent immune stimulation) which would inhibit extracellular TslA, how do authors envision TslA exerting its antibacterial activity.

Shedding of these proteins have been reported by some strains and under some growth conditions. However, it is not clear whether they are expressed or shed under conditions where we might anticipate bacterial antagonism to be active (for example during colonisation). Further work would be needed to investigate this in an appropriate colonisation model.

Perhaps the statement in line 302 that TsIA is required for bacterial antagonism is too strong since investigation of the physiological role of this toxin has not been exhausted, particularly it has not been explored using bacterial competition assays or other disease models.

We updated this text to soften our conclusion, also in response to comments from reviewer #2. The revised text (lines 311 - 325) now reads:

'Previous studies have reported a role for the *S. aureus* T7SS in murine abscess models of infection⁴⁵⁻⁴⁸. A strain lacking the *til1* cluster along with the *tsIA* cassette encoded on the ν Sa α island of USA300 also led to a significant reduction in bacterial burden in a mouse skin-infection model³⁷. The Til1 lipoproteins are known to be shed from the surface of wild type USA300 where they promote immune stimulation, which presumably accounts for the reduced virulence seen when the encoding genes are deleted^{36,37}. We therefore investigated whether TsIA contributed to virulence of *S. aureus* in the skin abscess model. As shown in Fig. 7, inactivation of the T7SS by deletion of *essC* resulted in a strong reduction in bacterial burden compared with the wild type USA300 strain. A strain lacking all *til1* genes was also significantly less virulent than the wild type, in agreement with prior observations³⁷. However, no statistically significant difference in bacterial burden was observed between USA300 and USA300 Δ *tsIA*, although there was some trend towards decreased virulence of the *tsIA* mutant in this infection model. (Fig. 7). It is possible that the virulence defect observed with the *essC* mutant is a cumulative effect resulting from the lack of secretion of TsIA alongside other T7SS toxins.'

Reviewer #4 (Remarks to the Author):

We are pleased to hear that *Nature Communications* supports ECRs in this way. No further response is required.

REVIEWERS' COMMENTS

Reviewer #2 (Remarks to the Author):

The authors have responded to previous critiques and I recommend acceptance.

Reviewer #4 (Remarks to the Author):
